# APOBEC3A is an oral cancer prognostic biomarker in Taiwanese carriers of an APOBEC deletion polymorphism

Ting-Wen Chen *et al.*#

Oral squamous cell carcinoma is a prominent cancer worldwide, particularly in Taiwan. By integrating omics analyses in 50 matched samples, we uncover in Taiwanese patients a predominant mutation signature associated with cytidine deaminase APOBEC, which correlates with the upregulation of *APOBEC3A* expression in the *APOBEC3* gene cluster at 22q13. *APOBEC3A* expression is significantly higher in tumors carrying *APOBEC3B*-deletion allele(s). High-level *APOBEC3A* expression is associated with better overall survival, especially among patients carrying *APOBEC3B*-deletion alleles, as examined in a second cohort ($n = 188$; $p = 0.004$). The frequency of *APOBEC3B*-deletion alleles is ~50% in 143 genotyped oral squamous cell carcinoma -Taiwan samples ($27A3B^{-/-}$:$89A3B^{+/-}$:$27A3B^{+/+}$), compared to the 5.8% found in 314 OSCC-TCGA samples. We thus report a frequent APOBEC mutational profile, which relates to a *APOBEC3B*-deletion germline polymorphism in Taiwanese oral squamous cell carcinoma that impacts expression of *APOBEC3A*, and is shown to be of clinical prognostic relevance. Our finding might be recapitulated by genomic studies in other cancer types.

#A full list of authors and their affliations appears at the end of the paper

Oral cancer is a common cancer worldwide. Approximately 300,373 new cases were diagnosed in 2012, making oral cancer a growing health concern[1, 2]. Oral squamous cell carcinoma (OSCC), which is the major subtype of oral cancer, accounts for >90% of all cases of oral cancer. In Taiwan, OSCC is a prevalent malignancy that represents the fourth most common cancer affecting males (M:F = 10.8:1; 6308:582 new cases in 2012, per the Taiwan Cancer Registry[3]). In addition to the known risk behaviors of cigarette smoking and alcohol drinking, Taiwanese men often indulge in the additional risk behavior of betel nut chewing[4, 5]. However, human papilloma virus (HPV) infection, which is another risk factor, is found at a much lower rate in Taiwan (~10%) compared with worldwide (~24 %)[6–8]. Despite the unique epidemiology of this disease, the contribution of the genetic background to its incidence or progression has not yet been explored in the Taiwanese population.

Genome-wide analyses of the head and neck cancer data reported by The Cancer Genome Atlas (TCGA)[9–11] and an India-based research group[12] revealed shared mutation spectrums, particularly in genes associated with cancer development (e.g., TP53, NOTCH1, FAT1, CASP8, and PIK3CA) and copy number variations (e.g., deletion of CDKN2A and amplification of FADD). These somatic mutations are likely to arise from distinct origins, and may thus generate unique signatures in combination[13, 14]. For head and neck cancer, three prevalent mutational signatures have been identified: age-associated signature 1B (~24.9%), APOBEC (Apolipoprotein B mRNA Editing enzyme, Catalytic polypeptide)-associated signature 2 (~35.2%), and smoking-associated signature 4 (~34.8%)[13].

The APOBEC genes encode members of a superfamily of cytidine deaminases that deaminate the cytidines of DNA and RNA to uracil, and have been implicated in diverse biological functions, including innate immunity and viral restriction[15–17]. The human genome encodes 11 APOBEC genes, seven of which (APOBEC3A, APOBEC3B, APOBEC3C, APOBEC3D, APOBEC3F, APOBEC3G, and APOBEC3H) form the APOBEC3 gene cluster at 22q13. A germline APOBEC3B (A3B) deletion polymorphism forms the fusion gene, APOBEC3A_B (A3A_B), which comprises the APOBEC3A (A3A) coding region and the APOBEC3B (A3B) 3′UTR. This polymorphism is rare in Africans and Europeans (0.9% and 6%, respectively) but common in East Asians, Amerindians, and Oceanics (36.9%, 57.7%, and 92.9%, respectively)[18]. Recent studies in TCGA tumors uncovered elevated expression of A3B in breast, bladder, cervix, lung (adenocarcinoma and squamous cell carcinoma), and head and neck cancers, and addressed the role of A3B in mediating genomic mutations[19, 20]. However, no significant correlation was found between the clinical outcome of breast cancer and the APOBEC3B-deletion polymorphism[19]. Both A3A and A3A_B were found to exhibit stronger cytidine deaminase activity than A3B in an in vitro study[21], and breast cancer patients harboring A3B deletions were found to have more APOBEC-dependent mutations than patients lacking an A3B deletion[22].

In this study, we extend our previous targeted sequencing effort in profiling OSCC mutations[23] to a comprehensive and systems-based interrogation of mutational landscapes and their pathological consequences in Taiwanese OSCC patients (OSCC-Taiwan). We carry out exomic and transcriptomic analyses of samples from a first cohort, and further evaluate our findings in a larger second cohort of patients for which we have complete clinical outcome information. These analyses allow us to compile a comprehensive mutational landscape for OSCC-Taiwan and identify novel associated mutations and alterations. In particular, we find that the APOBEC3B-deletion polymorphism is prevalent in the Taiwanese population, and that the corresponding expressional alteration is a key factor in determining the outcome

of OSCC in these patients. Taken together, the results from our integrated analyses provide insights into the genetic basis of disease-associated alterations of gene expression in OSCC, and suggest novel powerful biomarkers that may prove useful for precision medicine.

## Results

**Mutational landscape for OSCC-Taiwan.** To uncover genomic alterations that could prove useful as molecular markers for OSCC in Taiwan, we recruited patients who were ethnically Taiwanese and had been admitted to Chang Gung Memorial Hospital at Linkuo. We performed whole-exome sequencing (WES) of matched tumor/PBMC DNAs from 50 OSCC patients (75-Mbp target region, mean depth = 244 ± 54×), and RNA sequencing of matched tumor/adjacent normal tissues from 39 of the 50 patients (mean read count = 38.7 ± 15.2 million; Fig. 1a, Supplementary Fig. 1, Supplementary Tables 1, and Supplementary Data 1–2). A subset of the identified somatic mutations was validated in 49 tumor tissues using an IonAmpliSeq Comprehensive Cancer Panel (CCP; 409 cancer-related genes). We obtained an average coverage of 1165 ± 220×. One tumor sample was missed due to problems with DNA availability. A detailed workflow for our sequencing experiments and data analysis is shown in Supplementary Fig. 1.

In total, our exome sequencing analyses uncovered 24,051 somatic single-nucleotide variants (SNVs) and InDels (Supplementary Data 3 and Supplementary Fig. 2a). The targeted approach of CCP enabled us to independently validate 84.91% of the SNVs identified by our WES (Supplementary Data 4 and Supplementary Fig. 2b). We cross-referenced our data with that of 172 oral cavity cancer samples (OSCC-TCGA) from the Head-Neck Squamous Cell Carcinoma (HNSC) project of TCGA (HNSC-TCGA) and 106 oral cavity cancers from India (OSCC-India), as annotated in the International Cancer Genome Consortium (ICGC). We found that our patient cohort was characteristic of OSCC with regard to frequently mutated genes, such as TP53, FAT1, NOTCH1, PIK3CA, CDKN2A, and HRAS. Interestingly, however, our analysis also identified somatic mutations in genes that turned out to be unique (CENPV) or locally prevalent (DHRS4, RASA1, and SETD8; mutated in ≥ 10% of the Taiwanese patients) (Fig. 1b, Supplementary Table 2, and Supplementary Fig. 3). We further used our exome sequencing data to examine copy number alterations in our samples (Fig. 1c and Supplementary Fig. 4a). Our findings largely recapitulated the profile found in OSCC-TCGA (Supplementary Fig. 4b), which included significantly amplified/deleted regions encompassing genes such as EGFR, FGFR1, CCND1, FADD, FAT1, and CDKN2A (Supplementary Data 5 and Supplementary Data 6). However, we also uncovered new amplification peaks at 9p24.2 and 14q11.2, and new deletion peaks at 5q35.3, 6q21, 17q12, 12p13.31, 5q31.1, and 11p15.4 (Supplementary Fig. 4a vs. Supplementary Fig. 4b). A summary of the genomic mutational landscapes of all samples, including somatic mutations, copy number variations (CNVs), mutational signatures, and the correlations of these changes with each patient's clinical information, is presented in Fig. 1.

Comparative analysis of the transcriptomes of tumor and normal tissues allowed us to identify 3548 genes as being differentially expressed in OSCC-Taiwan; these included 1220 upregulated and 2328 downregulated genes (Supplementary Fig. 5). Notably, 91.3% of these differentially expressed genes (DEGs) were similarly identified in the OSCC-TCGA data set, indicating that the overall expression profile of OSCC is highly similar across ethnic groups. When we performed pathway analysis using MetaCore, we found that the identified DEGs were

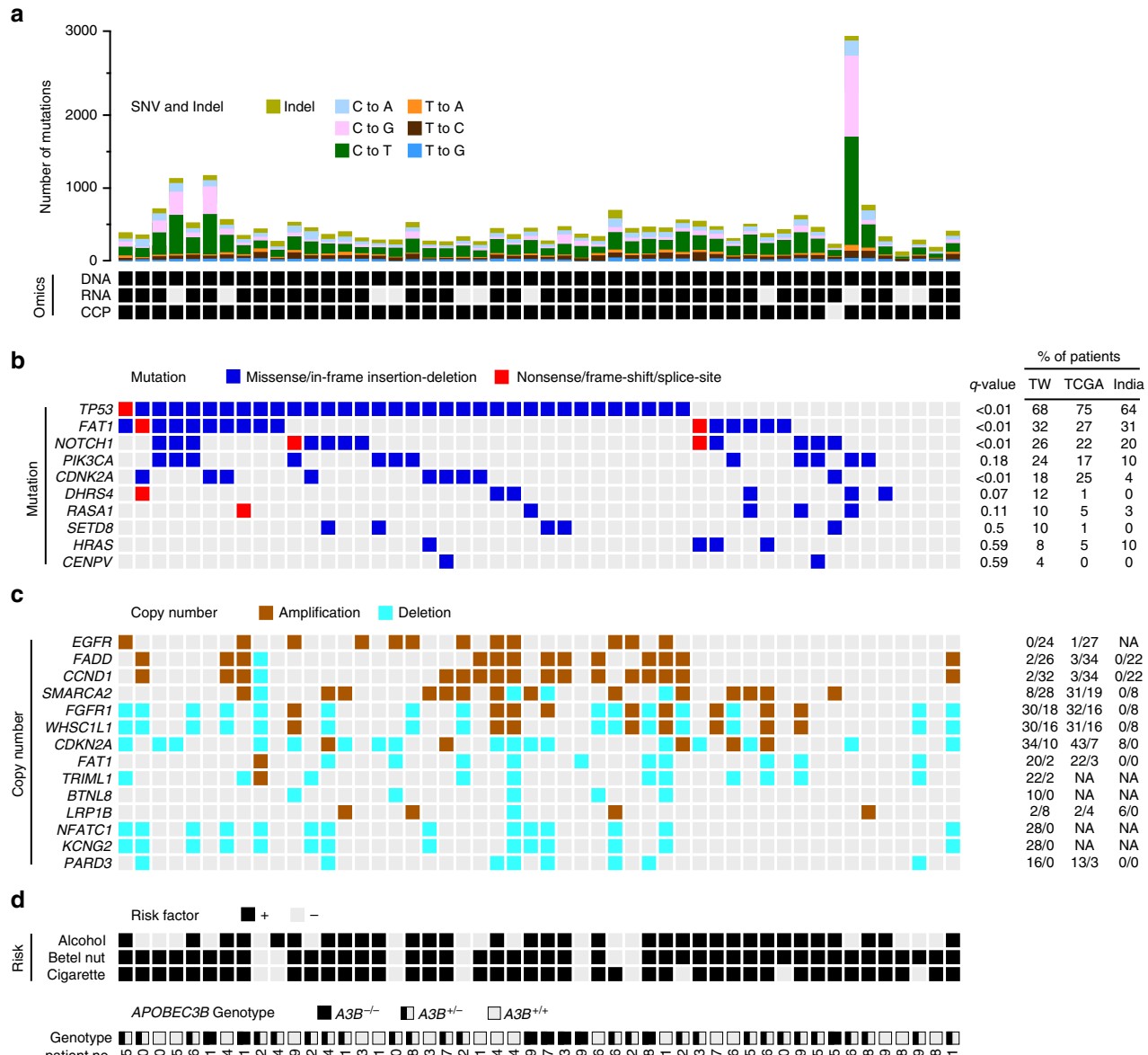

**Fig. 1** An integrated deep-sequencing approach identifies novel variant features underlying OSCC of a unique demographic origin. Summary data for the 50 OSCC-Taiwan cases. The four blocks correspond to the different types of data attributes. They represent, from *top* to *bottom*: **a** Mutation analyses in a series of 50 OSCC samples. The y-axis shows the number of mutation events and the omics data (*DNA* exome sequencing, *RNA* RNA-Seq, *CCP* comprehensive cancer panel), whereas the x-axis indicates the samples of the individual patients. **b** Heatmap representation of individual genes exhibiting somatic mutations in the 50 OSCC samples. The q-values (false discovery rates) represent the significance of each mutated gene, as determined using MutSigCV. **c** Heatmap representation of the copy number variations, compared with those from TCGA and India. SNVs were identified with Mutect, which applies a Bayesian classifier to detect mutations with allelic fractions of 0.1 or less ( <10%). For the number of mutation events, the mutation types are broken down by the indicated sequence features. For the mutation **b** and copy number analyses **c**, the tables on the right show the percentages of patients with the respective somatic sequence variation or amplification/deletion, as found in the OSCC-Taiwan (TW), OSCC-TCGA (TCGA), and OSCC-India (India) cohorts. *NA* data not available. **d** The risk exposure and *A3B* deletion genotypes. OSCC patients with the habits of alcohol, betel nut or cigarette are individually marked. For 3 APOBEC3B genotypes, *A3B⁻/⁻*, *A3B⁺/⁻*, and *A3B⁺/⁺* are shown with full, half and empty squares, respectively

enriched in pathways related to cell adhesion, cytoskeletal remodeling, and immune responses (Supplementary Data 7), providing hints into the pathological causes of OSCC.

**APOBEC mutational signatures correlate with elevated *A3A*.** Previous pan-cancer analyses in 7042 cancers of various types defined distinct mutational signatures as being genetic hallmarks of different cancer types[13, 14]. By applying a similar algorithm to data for the 20,963 somatic SNVs discovered herein, we identified

three mutational signatures as being enriched in OSCC-Taiwan: the APOBEC-associated signature (signature 2/13; 40%), the age-associated signature (signature 1A/1B; 37%), and the smoking-associated signature[8] (signature 5; 23%) (Figs. 2a and 2b). We then compared our data for these signatures with those from the 32 cancer projects archived in the TCGA/ICGC data portal, using the somatic SNVs identified in OSCC-TCGA as a reference. Notably, the APOBEC-associated signature was highly represented in the OSCC-Taiwan and OSCC-TCGA data sets, but to a lesser extent (17%) in the OSCC-India and HNSC-TCGA (31%)

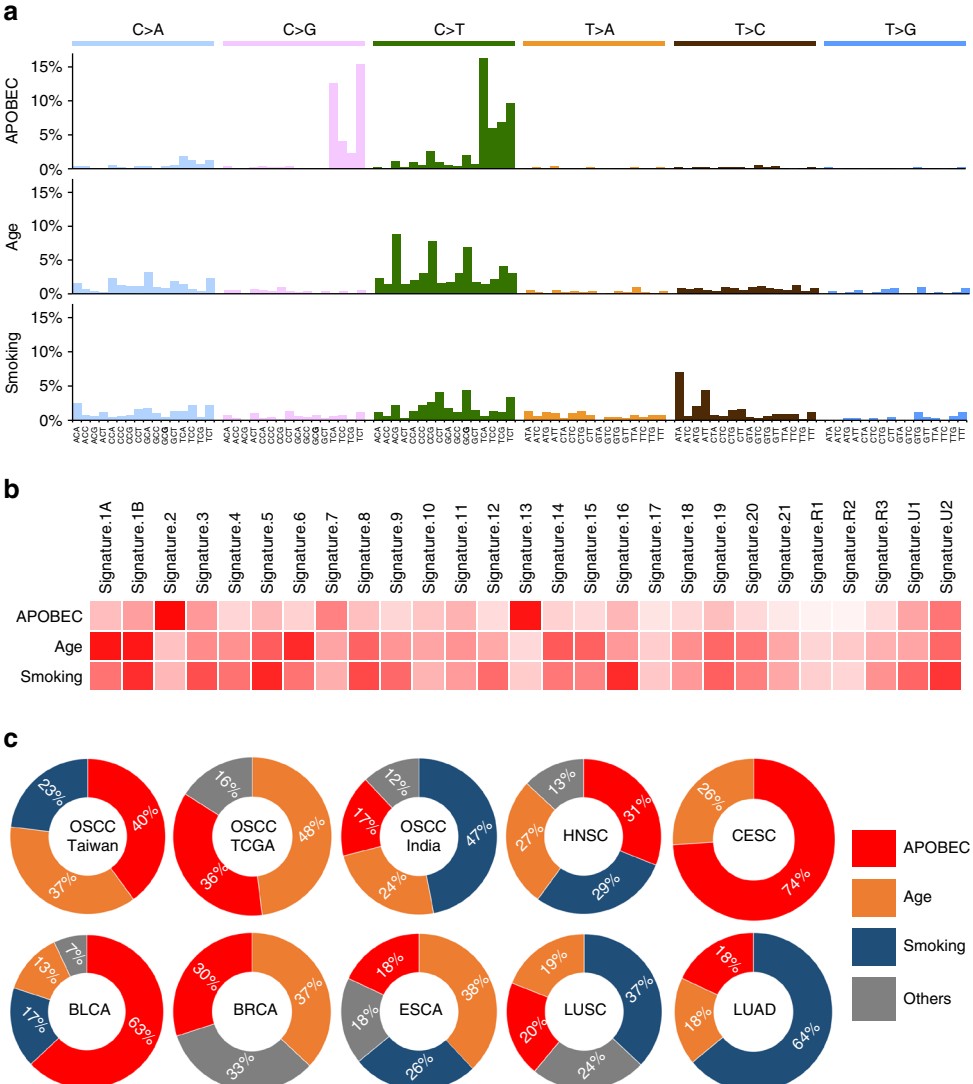

**Fig. 2** Mutational signature analysis of OSCC-Taiwan. **a** Three distinct mutational signatures were identified by our OSCC whole-exome sequencing. The spectra of base changes representing APOBEC (*signatures 2/13*), age (*signature 1*), and smoking (*signature 4/5*) are shown. The x-axis indicates the 96 combinations of trinucleotide motifs, while the y-axis represents the relative coefficient of the detected signature. **b** Heatmap of cosine-similarity results for the mutation spectrums of OSCC-Taiwan, coded by color. The cosine-similarity score, which ranges from 0 to 1, represents the extent of similarity to a particular signature. Among the 27 mutational signatures, the APOBEC, age, and smoking signatures are the most significant mutational signatures detected in OSCC samples (*dark red*). **c** The three overrepresented mutation signatures described in **a** were compared among datasets representing OSCC from Taiwan, India, and TCGA, as well as other TCGA tumor types carrying APOBEC-associated signatures. OSCC-TCGA is a subset of the HNSC (head and neck squamous cell carcinoma) data archived in TCGA. CESC, BLCA, BRCA, ESCA, LUSC, and LUAD correspond to cervical squamous cell carcinoma, bladder urothelial carcinoma, breast invasive carcinoma, esophageal carcinoma, lung squamous cell carcinoma, and lung adenocarcinoma, respectively

data sets (Fig. 2c and Supplementary Fig. 6). This suggests that the APOBEC-associated signature may be a predominant sequence feature in OSCC.

We next examined whether the observed overrepresentation of this mutation signature might be attributed to tumor-related alterations in *APOBEC* expression. As shown in Fig. 3a, our RNA-Seq results revealed that *A3A* and *A3B* were significantly elevated in tumors compared to adjacent normal tissues ($p < 0.0001$ for both genes), with *A3A* showing a considerably larger upregulation (the fold changes for *A3A* and *A3B* were 8.3 and 3.2, respectively). With respect to the other *APOBEC* genes, *A3G* was differentially elevated in tumors ($p = 0.02$), but its overall expression level was rather low. We further confirmed the significant upregulations of *A3A* and *A3B* ($p < 0.0001$) in a second cohort comprising 188 N/T paired OSCC samples (Supplementary Fig. 7 and Supplementary Table 3). Importantly,

we also detected enrichment of A3A-specific peptides in tumor proteomes, as assessed by iTRAQ-mass spectrometry (Fig. 3b and Supplementary Fig. 8).

To corroborate that *APOBEC* upregulation is linked with the above-described unique profile of genomic mutations, we analyzed the expression levels of *A3A* and *A3B* in cancer types previously reported to be enriched for mutation signatures 2 and 13[13]. We found that the expressions of *A3A* and *A3B* were generally elevated in these tumors, as previously reported[20, 24], and that OSCC exhibited the highest expression level of *A3A* among them (Fig. 3c). Of note, the expression levels of *A3A* and *A3B* were both significantly increased ($p < 0.001$) in the OSCC-TCGA dataset, with *A3B* showing the greater degree of elevation (Fig. 3c). Given that *A3A* and *A3B* are reportedly interferon (IFN)-inducible[25–29], this upregulation may reflect the activation of IFN signaling. Indeed, our transcriptome-sequencing data are

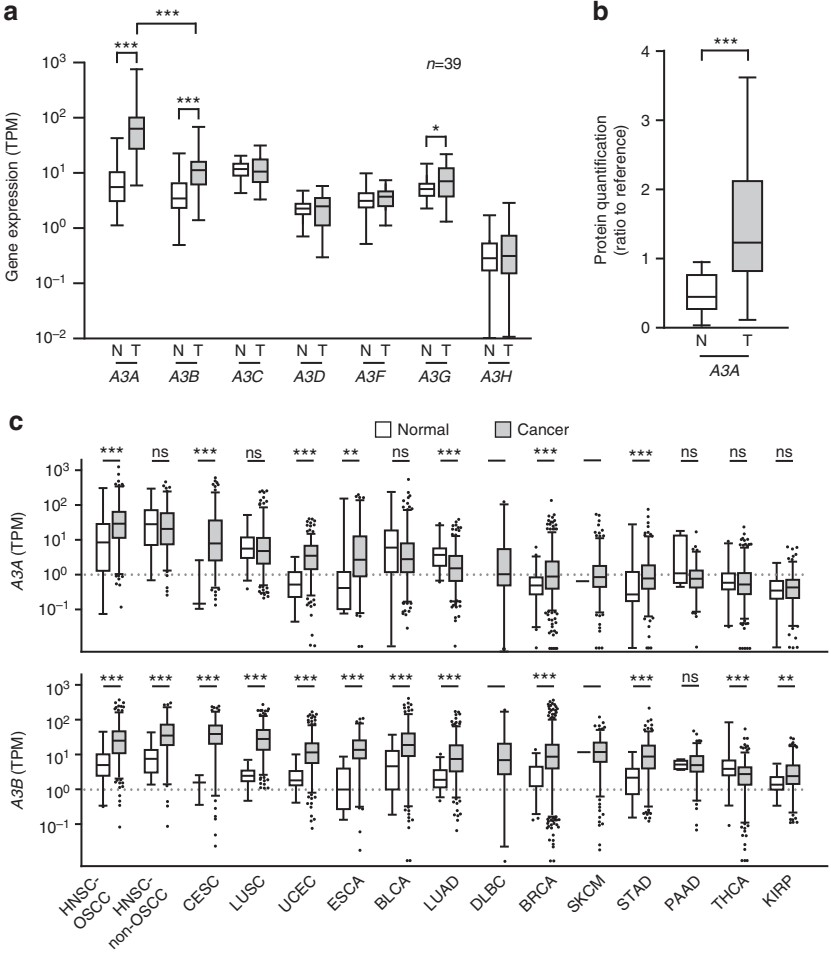

**Fig. 3** *APOBEC3* gene expression is altered in OSCC. **a** The mRNA expression levels of all seven *APOBEC3* genes in our paired OSCC samples were determined by RNA-Seq. The y-axis shows the TPM (transcripts per million) of each gene in paired tissue samples. N adjacent normal tissue; T tumor tissues. The expression profiles illustrate the tumor-specific up-regulations of *A3A*, *A3B* ($p < 0.0001$, Mann–Whitney *U*-test), and *A3G* ($p = 0.02$, Mann–Whitney *U*-test). In tumors, *A3A* was expressed at a higher level than *A3B* ($p < 0.001$). **b** A3A-specific peptides were identified in tissue proteomes using iTRAQ-mass spectrometry. The y axis represents the protein level relative to a common reference. A3A-specific peptides were significantly higher (~3 fold) in tumor tissues compared with corresponding normal tissues ($p < 0.001$, Mann-Whitney *U*-test). Of the 38 N/T paired tissues analyzed, we detected A3A peptides in 35 normal tissue samples and 36 tumor tissue samples. **c** Expression levels of *A3A* and *A3B* in different TCGA cancer types. Only cancers reported to have APOBEC-mutation signatures were included in this plot; they are ordered according to the *A3A* expression levels in their tumors. The y-axis shows the TPM of our RNA-Seq data for the *A3A* and *A3B* genes. The HNSC samples were further grouped into OSCC and 'others' (HNSC-non-OSCC). UCEC, DLBC, SKCM, STAD, PAAD, THCA, KIRP, CESC, BLCA, BRCA, ESCA, LUSC, and LUAD correspond to uterine corpus endometrial carcinoma, lymphoid neoplasm diffuse large B-cell lymphoma, skin cutaneous melanoma, stomach adenocarcinoma, pancreatic adenocarcinoma, thyroid carcinoma, kidney renal papillary cell carcinoma, cervical squamous cell carcinoma, bladder urothelial carcinoma, breast invasive carcinoma, esophageal carcinoma, lung squamous cell carcinoma and lung adenocarcinoma, respectively. Box plots show the distribution of expression of indicated APOBEC genes. Boxes extend from the third (Q3) to the first (Q1) quartile, with the line at the median; whiskers extend to 2.5 and 97.5 percentiles

consistent with this hypothesis (Supplementary Fig. 9 and Supplementary Data 7). Furthermore, we identified several miRNAs that are downregulated in OSCC (from the TCGA data) and were predicted by TargetScan to target the *A3B* 3′UTR[30]. We examined the regulatory role of one of them, miR-409, and found that it reduced the activity of a luciferase reporter gene carrying the *A3B* 3′UTR (Supplementary Fig. 10).

The accumulation of APOBEC-associated mutations requires inactivation of the DNA repair system (i.e., *TP53* mutation), as reported in breast cancer[31]. Here, we discovered that the amounts of APOBEC-signature mutations were significantly correlated with the transcript abundance of *A3A*, but not *A3B*, in 27 OSCC patients with mutation of *TP53* ($p = 0.038$ and 0.239, respectively; Supplementary Fig. 11). As the hypermutation signature of *A3A* may be distinguished from that of *A3B* (YTCA vs. RTCA)[32], we

also examined their relative incidences in our samples. We found that the YTCA:RTCA and YTCW:RTCW ratios were both about 7:3, and that the value of YTCA/YTCW was significantly higher than that of RTCA/RTCW (Wilcoxon rank sum test, $p = 0.0009$ and 1.704e-07, respectively; Supplementary Table 4), pinpointing *A3A* as the major mutator contributing to the APOBEC-associated signature in OSCC-Taiwan. Four out of the 50 cases were found to be HPV DNA-positive. However, three of them carried *TP53* mutations and all four were E6 transcript-negative (Supplementary Table 5)[33–35], suggesting that the mutation signatures were unrelated to HPV infection. Viewed together, the results from our integrated sequencing and expression analyses are in line with a model in which the up-regulation of *A3A* activity in OSCC-Taiwan contributes to the presence of a dominant mutation signature.

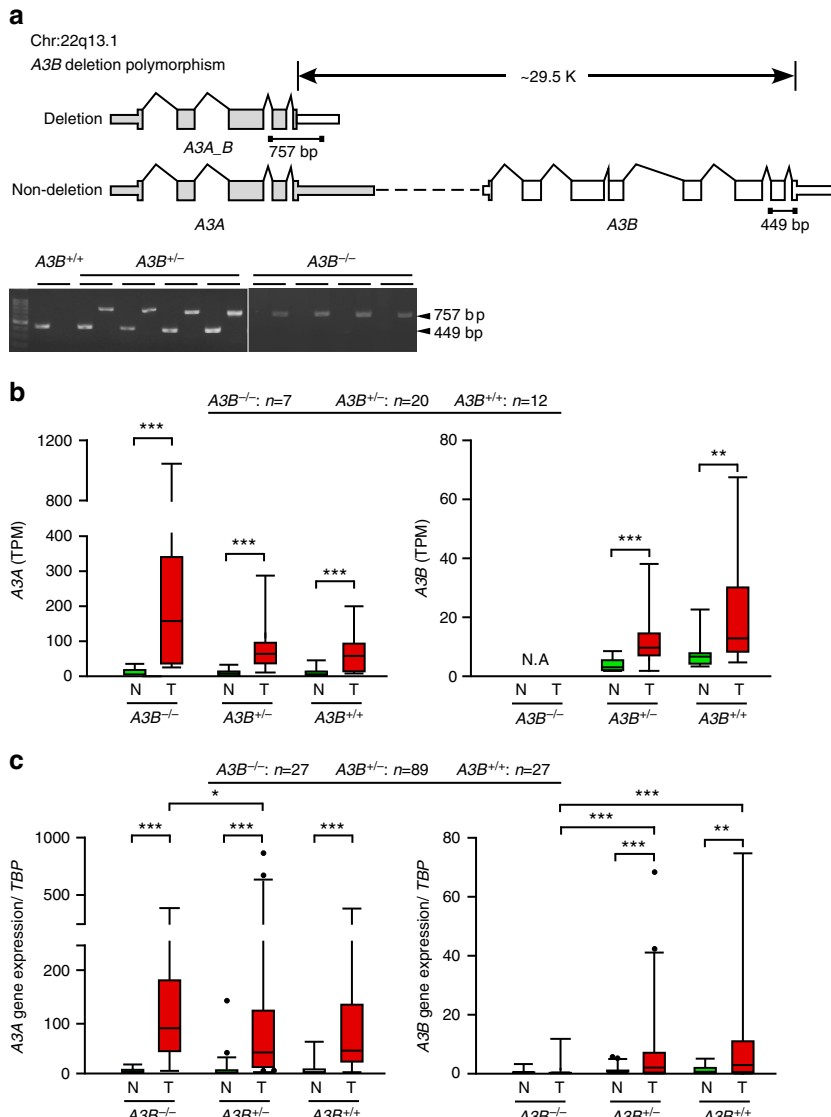

**Fig. 4** The deletion polymorphism of the *A3A-A3B* genomic locus and upregulation of *A3A* and *A3B* in OSCC-Taiwan. **a** Schematic depiction of the gene structures and genomic organizations of the deletion polymorphism (*top*) and non-deleted (*bottom*) versions of the *APOBEC3A-APOBEC3B* genomic locus. The deletion variant (*APOBEC3A_B* genotype) arises from a 29.5-kb genomic deletion spanning from the 3′UTR of *A3A* to the eighth exon of *A3B*. PCR-based genotyping analysis was used to distinguish non-deletion and deletion alleles according to the size of the amplified product (449 and 757 bp, respectively). *A3B*⁺/⁺, *A3B*⁺/⁻, and *A3B*⁻/⁻ represent non-carrier individuals and those heterozygous and homozygous for the deletion allele, respectively. **b** Genotype-biased expressional alterations of *A3A* and *A3B* in OSCC. Based on RNA-Seq-determined TPM values (*y*-axis), the mRNA expression levels of *A3A* (*left*) and *A3B* (*right*) were determined in the initial cohort of 39 paired samples (N, adjacent normal tissues; T, tumor). Patients are grouped according to their *APOBEC3B*-deletion genotypes, with the number of patients in each group (*n*) indicated at the top. **c** Genotype-specific relative expression of *A3A* in tumor (T) vs. normal tissue (N) samples, as determined by RT-PCR. The *y*-axis represents the expression level of *A3A* (or *A3B*) relative to that of *TBP* (TATA binding protein gene). Individuals of the *A3B*⁻/⁻ genotype exhibited a greater tumor-specific upregulation of *A3A* compared with those of the *A3B*⁺/⁻ genotype ($p = 0.0354$) (*$p < 0.05$; **$p < 0.001$; ***$p < 0.0001$, Wilcoxon signed-rank test). Box plots show the distribution of expression of indicated *APOBEC* genes. Boxes extend from the third (Q3) to the first (Q1) quartile, with the line at the median; whiskers extend to 2.5 and 97.5 percentiles

**Genotyping and expression profiling of the *APOBEC* locus.** A previous report described a germline deletion polymorphism at the *APOBEC3* locus that removes the coding region of *A3B*, leading to expression of a variant transcript in which the *A3A* coding sequence is fused to the 3′UTR of *A3B*[21] (termed *A3A_B*) (Fig. 4a). In our 50 OSCC matched samples, we detected germline deletion of the *A3B* coding sequence in 33 individuals: eight *A3B*⁻/⁻ individuals (homozygous for the deletion allele) and 25 *A3B*⁺/⁻ individuals (heterozygous for the deletion allele) (Supplementary Fig. 12a). We confirmed our exome sequencing results by PCR using genotype-specific primer sets that

distinguished among the *A3B*⁻/⁻ (a 757-bp PCR product), *A3B*⁺/⁻ (449-bp and 757-bp), and *A3B*⁺/⁺ (449-bp) genotypes (Fig. 4a). Our RNA-Seq data further substantiated the presence of a variant transcript corresponding to this genomic polymorphism (Supplementary Data 8), and RT-PCR analysis with fusion-sensitive primers independently confirmed the expression of the deletion variant in *A3B*⁻/⁻ and *A3B*⁺/⁻ samples (Supplementary Fig. 12b).

Next, we used our RNA-Seq data to analyze the correlation between the various genotypes and the expression levels of *A3A* and *A3B*. As shown in Fig. 4b and Supplementary Fig. 13, *A3A* expression was significantly elevated in tumor tissues compared

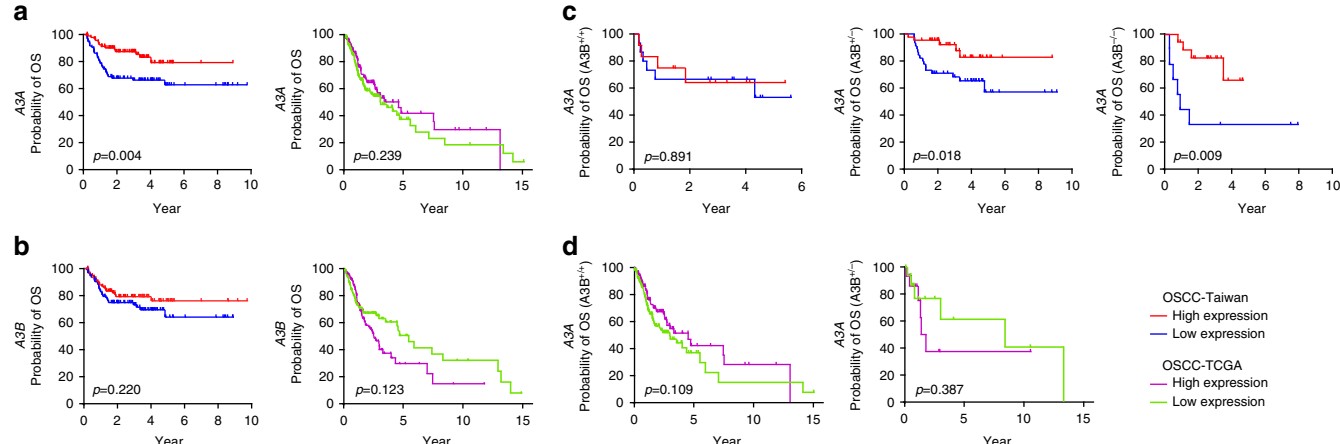

**Fig. 5** Kaplan–Meier plot for overall survival (OS) by *A3A* and *A3B* expression and genotype in OSCC-Taiwan and OSCC-TCGA. **a** Kaplan–Meier plot showing that the 10-year OS rates for patient subgroups stratified by high vs. low *A3A* expression were 85.1% and 65.9%, respectively ($p = 0.004$) in the OSCC-Taiwan data set, whereas no significant difference was found in the OSCC-TCGA data set ($p = 0.239$). **b** No significant difference in OS was found for subgroups stratified according to *A3B* expression (*right*) in the OSCC-Taiwan and OSCC-TCGA datasets. **c** Kaplan–Meier plot for OS in subgroups stratified by *A3A* expression among the 143 patients in the OSCC-Taiwan data set. High *A3A* in patients with the $A3B^{+/-}$ ($n = 89$) and $A3B^{-/-}$ ($n = 27$) genotypes, but not the $A3B^{+/+}$ ($n = 27$) genotype, was significantly correlated with better OS. **d** Kaplan–Meier plot for OS in subgroups stratified by *A3A* expression among the 312 patients in the OSCC-TCGA data set. No significant correlation was found in patients with the $A3B^{+/+}$ ($n = 278$) or $A3B^{+/-}$ ($n = 34$) genotypes. The *y*-axis shows the probability of OS according to high and low *A3A* expression. The survival rate was estimated by Kaplan–Meier plotting and compared by log-rank test; all *p*-values are two-sided, with the significance level set at $p < 0.05$

**Table 1 Multivariate analysis of overall survival (OS) or disease specific survival (DSS) in the second cohort ($n = 188$) after treatment**

| Characteristics | Hazards ratio (OS) (95% confidence interval) | *p*-value (OS) | Hazards ratio (DSS) (95% confidence interval) | *p*-value (DSS) |
|---|---|---|---|---|
| *Age (median: 50.6)* | | | | |
| < 50.6y | 1.000 (Reference) | 0.321 | 1.000 (Reference) | 0.384 |
| > 50.6y | 1.351 (0.745–2.451) | | 1.342 (0.692–2.602) | |
| *Gender* | | | | |
| Male | 1.000 (Reference) | 0.459 | 1.000 (Reference) | 0.206 |
| Female | 0.685 (0.252–1.864) | | 0.517 (0.186–1.440) | |
| *Overall pathological stage* | | | | |
| I–II | 1.000 (Reference) | 0.022[a] | 1.000 (Reference) | 0.079 |
| III–IV | 3.128 (1.172–8.344) | | 2.460 (0.900–6.723) | |
| *Perineural invasion* | | | | |
| No | 1.000 (Reference) | 0.058 | 1.000 (Reference) | 0.018[a] |
| Yes | 1.823 (0.979–3.395) | | 2.329 (1.150–4.716) | |
| *Cell differentiation*[b] | | | | |
| W-D + M-D P-D | 1.000 (Reference) 1.035 (0.456–2.348) | 0.935 | 1.000 (Reference) 0.857 (0.332–2.215) | 0.750 |
| *Bone invasion* | | | | |
| No | 1.000 (Reference) | 0.901 | 1.000 (Reference) | 0.634 |
| Yes | 1.042 (0.543–1.999) | | 0.831 (0.389–1.779) | |
| *APOBEC3A expression level* | | | | |
| Low | 1.000 (Reference) | 0.035[a] | 1.000 (Reference) | 0.028[a] |
| High | 0.499 (0.261–0.954) | | 0.444 (0.215–0.917) | |

Multivariate analyses also adjusted with patients' age and sex
[a] statistically significant
[b] W-D: well-differentiated, M-D: moderately-differentiated, and P-D: poorly-differentiated, squamous cell carcinoma

to adjacent normal tissues from all three genotypes; moreover, the level was about 2-fold higher in carriers of the $A3B^{-/-}$ genotype compared to those harboring the $A3B^{+/-}$ and $A3B^{+/+}$ genotypes. The expression of *A3B* was also significantly elevated in tumor tissues from patients of all three genotypes, but its expression was about 5- to 20-fold lower than that of *A3A*. Next, we examined the prevalence and expressional consequence of this deletion

polymorphism in the second cohort (143 of 188 individuals were genotyped: $A3B^{-/-}$, $n = 27$; $A3B^{+/-}$, $n = 89$; and $A3B^{+/+}$, $n = 27$). Among these individuals, *A3A* expression was significantly elevated in all tumor tissues, with higher levels seen in $A3B^{-/-}$ individuals vs. those with the $A3B^{+/-}$ or $A3B^{+/+}$ genotypes (Fig. 4c, *left*). In contrast, the expression levels of *A3B* were higher in the tumor tissues compared to normal tissues of $A3B^{+/-}$ and

**Table 2 The clinicopathological characteristics related to the expression of *APOBEC3A* in the second cohort (*n* = 188)**

| Patient categories | Case No. | APOBEC3A expression level | | p-value |
|---|---|---|---|---|
| | | Low (%) | High (%) | |
| *Gender* | | | | |
| Male | 172 | 87 (46.2) | 85 (45.2) | 0.794 |
| Female | 16 | 7 (3.72) | 9 (4.79) | |
| *Age*[a] | | 51.2 ± 10.8 (80, 29) | 52.4 ± 10.0 (79, 30) | 0.327 |
| *pT status* | | | | |
| 1–2 | 87 | 39 (20.7) | 49 (26.1) | 0.109 |
| 3–4 | 100 | 56 (29.8) | 44 (23.4) | |
| *pN status* | | | | |
| (−) | 97 | 42 (22.3) | 55 (29.2) | 0.079 |
| (+) | 91 | 52 (27.6) | 39 (20.7) | |
| *Overall pathological stage* | | | | |
| I–II | 56 | 19 (10.1) | 37 (19.6) | 0.006[b] |
| III–IV | 132 | 75 (39.8) | 57 (30.3) | |
| *ECS*[c] | | | | |
| (−) | 142 | 68 (36.1) | 74 (39.3) | 0.396 |
| (+) | 46 | 26 (13.8) | 20 (10.6) | |
| *Cell differentiation*[d] | | | | |
| W-D + M-D | 163 | 78 (41.4) | 85 (45.2) | 0.197 |
| P-D | 25 | 16 (8.5) | 9 (4.7) | |
| *Perineural invasion* | | | | |
| No | 103 | 44 (23.4) | 59 (31.3) | 0.027[b] |
| Yes | 85 | 50 (26.6) | 35 (18.6) | |
| *Second primary occurrence* | | | | |
| No | 174 | 87 (46.2) | 87 (46.2) | 1.000 |
| Yes | 14 | 7 (3.72) | 7 (3.72) | |
| *Bone invasion* | | | | |
| No | 137 | 61 (32.4) | 76 (40.4) | 0.013[b] |
| Yes | 51 | 33 (17.5) | 18 (9.57) | |
| *Tumor depth*[a] | | 15.1 ± 10.0 (48, 1) | 14.9 ± 11.1 (55, 2) | 0.625 |

[a]Mean ± SD, median (maximum, minimum)
[b]Statistically significant
[c]ECS: extracapsular spread
[d]W-D: well-differentiated, M-D: moderately-differentiated, and P-D: poorly-differentiated, squamous cell carcinoma

$A3B^{+/+}$ genotype individuals, whereas it was not detected in the tumors of $A3B^{−/−}$ genotype individuals, as expected (Fig. 4c, right). We also analyzed data from 313 TCGA oral cancer samples predominantly obtained from the US; the cases included 278 $A3B^{+/+}$ genotype individuals, 34 $A3B^{+/−}$ individuals, and 1 $A3B^{−/−}$ individual[22]. In this dataset, we failed to detect any significant difference in *A3A* expression between $A3B^{+/−}$ and $A3B^{+/+}$ individuals, while the level of *A3B* was modestly higher in $A3B^{+/+}$ than $A3B^{+/−}$ individuals (Mann–Whitney test, *p* = 0.756 and 0.014, respectively; Supplementary Fig. 14). Viewed together, these results strongly indicate that the differential expression of *A3A* and *A3B* in OSCC-Taiwan could be attributed in part to the genetic background of these patients.

**A3A as a key clinicopathological determinant in OSCC.** Expressional data obtained from qRT-PCR were used to stratify the patients of our second cohort (*n* = 188; Supplementary Table 3) into high vs. low expression groups, using the median as the cutoff value. We then tested whether *A3A* expression was associated with overall survival (OS). Kaplan–Meier survival analysis revealed that the 10-year OS rates were 85.1% and 65.9% for the high and low *A3A* expression subgroups, respectively, and were significantly different based on the log-rank test (*p* = 0.004; Fig. 5a, *left*). We did not find any significant clinical correlation for *A3A* expression in the OSCC-TCGA dataset (*p* = 0.239; Fig. 5a, *right*). The 10-year disease-specific-survival (DSS) rates were also significantly different between the high and low expression groups in the second cohort (85.1% and 65.9%, respectively; *p* = 0.004 by log-rank test). A multivariate analysis was carried out using age, sex, overall stage, perineural invasion, tumor differentiation, bone invasion, and *A3A* expression. The results demonstrated that higher overall stage and low *A3A* expression were associated with poorer prognosis for both OS and DSS, indicating that these represent independent factors for prognostic prediction following OSCC treatment (Table 1). In contrast, the expression of *A3B* did not correlate with either of the 10-year survival rates (Fig. 5b). Conversely, no significant correlation between OS and the expression of *A3A* or *A3B* was found in the OSCC-TCGA dataset (Figs. 5a, *right*, and 5b, *right*).

Given the putative clinical relevance of the *A3A* alterations identified in our samples, we evaluated the relationships between *A3A* expression and various clinicopathological characteristics of the OSCC patients. We found that lower *A3A* expression was significantly associated with advanced overall stage, positive perineural invasion, and bone invasion (*p* = 0.006, 0.027, and 0.013, respectively; Table 2). However, we failed to establish any association between *A3A* expression in OSCC tumors and patient age, sex, pT status, pN status, extracapsular spread, differentiation, second primary occurrence, or tumor depth. Of the 188 patients with clinical outcome data, we were able to genotype 143 (Fig. 4c). We used these data to evaluate the association of *A3A* expression and survival among patients grouped by genotype. As shown in Fig. 5c and Supplementary Fig. 15, high *A3A* expression in the $A3B^{+/−}$ and $A3B^{−/−}$ genotypes was associated with better OS (*p* = 0.018 and 0.009, respectively; Fig. 5c), DSS (*p* = 0.010 and 0.009, respectively; Supplementary Fig. 15b and Supplementary Fig. 15c), and disease-free survival (DFS; *p* = 0.016 and 0.001, respectively; Supplementary Fig. 15b and Supplementary Fig. 15c). In contrast, the *A3A* level was not associated with survival among individuals of the $A3B^{+/+}$ genotype (*p* = 0.891, 0.200, and 0.998 for OS, DSS, and DFS, respectively; Fig. 5c and Supplementary Fig. 15a). Our results indicated that high *A3A* predicts better OS, DSS, and DFS, whereas *A3B* expression was not significantly correlated with treatment outcomes in our population. Of note, we did not find such a significant clinical correlation for *A3A* expression levels in $A3B^{+/+}$ or $A3B^{+/−}$ individuals from the OSCC-TCGA dataset (Fig. 5d). Considering that the OSCC-TCGA data exhibited an overall less pronounced *A3A* upregulation together with much less frequent *A3B* deletion (5.8%), our results further suggest that the high *A3A* expression in our overall dataset arose mainly from the $A3B^{+/−}$ and $A3B^{−/−}$ genotypes, which were prominent in OSCC-Taiwan but not OSCC-TCGA.

**Discussion**

To our knowledge, this is the first study to explore the physiological manifestation of constitutional variants among *APOBEC* genes in OSCC. Although our sample size was limited, our integration of exomic and RNA sequencing data from matched samples, combined with the validation of our findings in a larger cohort of clinical samples, provides a complementary breadth and depth of molecular information. This strategy enabled us to uncover several key novel attributes of OSCC, as follows: (1) This is the first report of *APOBEC3B*-deletion polymorphisms in a Taiwanese population, comprised mainly of Han-Chinese. Our

analysis reveals that allele deletion is frequent in this population (~50% in 143 genotyped patients), emphasizing that ethnicity is key to the genetic basis for this disease. (2) The constitutional deletion of both alleles of the *A3B* locus (*A3B*$^{-/-}$ genotype) is associated with the highest level of *A3A* expression, which may arise from altered 3′UTR regulation and enhanced IFN signaling in OSCC. (3) High-level *A3A* expression is correlated with better outcomes, particularly for patients carrying *APOBEC3B*-deletion alleles. (4) High-level *A3A* expression may serve as an independent prognostic biomarker for OSCC. Our findings indicate that the *APOBEC3B*-deletion genotype is strongly associated with *A3A* expression and clinical outcome. Given that APOBEC-induced mutations are now known to be prevalent in many cancers, it will be worthwhile to test this association in a larger cohort or in other cancer types in the future.

Our RNA-Seq data indicate that the transcript level of *A3A* is generally at least 10-fold higher than that of *A3B* in OSCC tumors, particularly among *A3B*$^{-/-}$ genotype individuals. Notably, and in contrast to this tumor-associated up-regulation, the *A3A* expression levels in normal tissues of all three genotypes were low. This difference is likely to reflect two major tumor-intrinsic alterations. First, *APOBEC* exhibits IFN inducibility[25–29], which is consistent with its role in antiviral innate immunity[36]. Enhanced IFN signaling may thus trigger the upregulation of *A3A*. Indeed, heightened expression of genes in the IFN signaling pathway was evident in our RNA-Seq data. Moreover, a previous proteomic analysis of OSCC also showed that the IFN pathway is enriched at this level in tumor tissues[37]. Second, as *A3A* was highly expressed in the tumors of individuals carrying the *A3B*$^{-/-}$ and *A3B*$^{+/-}$ genotypes, these polymorphisms may be linked *in cis* to the elevated expression of *A3A*. However, it is not yet known how this genomic disruption might alter *A3A* expression. A recent report found that the transcript for the fusion gene, *A3A_B*, which links the coding region of *A3A* to the 3′UTR of *A3B*, was more stable than the *A3A* transcript[21], suggesting that miRNAs may be involved. To address this possibility, we used sequence-prediction tools[30] to identify several candidate miRNAs with *A3B* 3′UTR-targeting potential. Interestingly, most of these miRNAs were downregulated in the oral cavity cancer subset of the TCGA dataset. Among the identified candidates, we validated the suppressive effect of one, miR-409, using a 3′UTR reporter assay. We found that overexpression of miR-409 reduced reporter activity driven by the *A3B* 3′UTR by 20% but did not affect that driven by the *A3A* 3′UTR (Supplementary Fig. 10). Collectively, our observations are consistent with the notion that, among patients carrying the *APOBEC3B*-deletion polymorphism, the 3′UTR of the variant *A3A_B* transcript may confer a new layer of regulation in OSCC that depends on both miRNAs and the disease context.

Many quantification methods utilize expectation-maximization algorithms to estimate the maximum likelihood expression levels for multiple mapped reads. Special care should be taken, however, when using RNA-Seq data to quantify the expression levels of *A3A*, *A3A_B*, and *A3B* in samples carrying the *APOBEC3B*-deletion genotype. As *A3A_B* and *A3B* are identical in their 3′ UTR sequences, the reads for *A3A_B* could be mis-mapped to *A3B* and wind up being assigned to *A3B* in *A3B*$^{-/-}$ samples. As *A3A_B* is not presently annotated in the human genome references, it may not have been considered in previous analyses involving transcriptome quantification. Thus, our findings further emphasize that it is important to cross-reference expressional data with genomic information.

Intriguingly, we found that high *A3A* expression was associated with better OS of our OSCC patients, suggesting that the upregulation of *A3A* may impact tumors. As most of the sampled patients had been subjected to therapy, however, this seemingly conflicting finding may be explained in the context of treatment. Anti-cancer therapeutics may create an environment of ongoing DNA damage, thereby producing the single-stranded (ss) DNAs that act as substrates for the APOBEC enzymes. The enhancement of *A3A* activity in OSCC tumor cells may accelerate the removal of damaged tumor cells, thereby increasing the efficacy of chemotherapy and improving the treatment response[38–40]. Notably, we failed to find any significant association of *A3A* or *A3B* expression with OS for the 314 samples archived in TCGA (Fig. 5a, *right*). This may imply the impact of genetic variations on Taiwanese OSCC patients in comparison with patients from other regions, which has not been reported before. Thus, a patient's racial background should be considered during clinical decision-making and therapy, as a key practice of precision medicine.

Betel nut chewing is a regional risk factor for OSCC, and 40 out of the 50 OSCC patients recruited in the discovery cohort had a history of both betel nut chewing and cigarette smoking. To explore the possible impact of betel nut chewing on the genomics of OSCC in Taiwan, we analyzed the mutational profiles and expression levels of 24 genes previously linked to betel nut chewing[5]. We did not find any strongly recurrent mutation among these genes in the tumor samples, but all of the tested genes were differentially expressed in the tumors compared with adjacent normal tissues. Our findings of altered expression levels were consistent with those reported in previous studies, and suggest that betel nut use may elicit certain cellular changes. The overwhelming majority (80–90%) of our study subjects in both cohorts reported chewing betel nuts. Thus, we cannot use the present data to establish any link between betel nut chewing and the expression levels of APOBECs. Future investigations that include sufficient non-betel nut chewers are warranted to examine the effect of this carcinogen on these important proteins.

## Methods

**Patient characteristics and clinical specimens**. Two cohorts of patients who are ethnically Taiwanese were enrolled in Chang Gung Memorial Hospital at Linkuo. The first cohort comprised 50 treatment-naive OSCC patients who underwent comprehensive preoperative work-ups, and curative resection with/without postoperative adjuvant therapy. Sections of tumors that contained at least 75% tumor nuclei among the total cellular nuclei were used for DNA and RNA isolation and quantification before subjected Next Generation Sequencing experiments. Data on demography, risk exposure, clinical characteristics, and histopathological features were collected and are provided in Supplementary Table 1. For the second cohort, tumor specimens were obtained from 188 consecutively enrolled patients (172 men and 16 women) between June 2006 and November 2014, and all had regular follow-up information. Patient characteristics are presented in Supplementary Table 3. Institutional research ethical approval and written informed consent were obtained for all participants in the study (Institutional Review Board of Chang Gung Memorial Hospital, Taiwan). In general, patients underwent standard preoperative work-ups according to institutional guidelines, including a detailed medical history, a complete physical examination, computed tomography or magnetic resonance imaging scans of the head and neck, chest radiographs, a bone scan, and an abdominal ultrasound. Primary tumors were excised with adequate margins under intraoperative frozen-section control. Pathological TNM classification of all tumors was established according to the American Joint Committee on Cancer Staging Manual (2010). After discharge, all patients had regular follow-up visits every 2 months for the first year, every 3 months for the second year, and every 6 months thereafter.

The OSCC primary tumors were all excised with adequate surgical margins, and the tumor margin tissue was sent for intraoperative fresh frozen section histopathology analysis. If margins were not deemed to be tumor free, further resection was performed. Nonetheless, five of the 188 resected OSCC specimens were found to be tumor-positive in surgical margins. Various types of neck dissection were performed according to the primary tumor site and clinical lymph node status. Postoperative radiotherapy was performed on patients with pathologically identified T4 tumors and positive lymph nodes within 6 weeks following surgery. Patients with any of the following pathological features received adjuvant concurrent chemoradiotherapy: metastasis in multiple neck lymph nodes, extracapsular spread, positive surgical margins, perineural invasion or nodal dissemination at level 4 or 5. The chemotherapy was a cisplatin-based regimen, and the total radiation dose was 66 Gy. The prescribed dose was delivered in fractions of 1.8–2 Gy per day for 5 days per week.

Patients underwent standard postoperative work-ups according to institutional guidelines, which included complete physical examination at regular follow-up visits. Computed tomography or magnetic resonance imaging of head and neck and chest radiographs were performed 3 months after the treatment and every 6 months for 3 years. Additional radiological examinations, bone scans, and abdominal ultrasonography were performed for any suspicious recurrence or second primary tumors noted clinically. All patients completed regular follow-up visits every 2–3 months for the 1st year after discharge, every 3–4 months for the second and third years, and every 6 months thereafter.

**Detection of human papillomavirus by E6 nested PCR**. For tissue samples, genomic DNA was extracted with a DNeasy Blood & Tissue kit (Qiagen, Chatsworth, CA, USA) according to the manufacturer's instructions. HPV detection and typing were performed as described previously[41]. In brief, 50 ng of genomic DNA was used to perform the first-round PCR with GP-E6/E7 consensus primers (GP-E6-3F: 5′-GGG WGK KAC TGA AAT CGG T -3′; GP-E7-5R: 5′- CTG AGC TGT CAR NTA ATT GCT CA -3′; GP-E7-6R: 5′- TCC TCT GAG TYG YCT AAT TGC TC-3′) to facilitate initial amplification of the genomic DNA of all known mucosal HPV genotypes and provide enough material to be re-amplified in nested PCRs with type-specific primers. The product size of GP-E6/E7 is 603 ~ 630 bp. Nested amplification of the GP-E6/E7 PCR products with type 16- and 18-specific primers (Nested HPV16-F: 5′- CAT ATA TTC ATG CAA TGT AGG TGT A - 3′ and Nested HPV16-R: 5′- CAC AGT TAT GCA CAG AGC TGC - 3′; Nested HPV18-F: 5′- GTT GTG AAA TCG TCG TTT TTC A -3′ and Nested HPV18-R: 5′- CAC TTC ACT GCA AGA CAT AGA - 3′) was chosen to achieve exact typing of the HPV infections. The product sizes of HPV 16 and 18 were 457 and 322 bp, respectively. All positive PCR products were confirmed by sequencing analysis.

**Genomic DNA extraction**. Genomic DNA was extracted from the tumor tissues using a QIAamp DNA Mini kit (Qiagen), and from whole-blood samples using a Puregene Blood Core kit (Qiagen), following the manufacturer's protocols. The quality and DNA concentrations of the samples were assessed using a Qubit fluorometer (Thermo Fisher Scientific) and 0.8% agarose gel electrophoresis.

**Next generation sequencing**. For whole-exome sequencing, high-quality genomic DNA (1.5 μg per sample) was subjected to capture using a SureSelect Human All Exon v5 + UTR kit according to the manufacturer's protocol (Agilent Technologies). The qualified exome-captured libraries were sequenced using a HiSeq 2000 with the TruSeq PE Cluster kit v3 and TruSeq SBS kit v3 (all from Illumina) according to the manufacturer's protocol. For RNA sequencing, libraries were prepared using the TruSeq RNA Access Library Prep Guide (Part # 15049525 Rev. B; Illumina) according to the manufacturer's instructions. Equal concentrations of each library were sequenced using a NextSeq 500 (Illumina) platform. For target sequencing, libraries were prepared from each sample using an Ion AmpliSeq Comprehensive Cancer Panel (CCP; Thermo Fisher Scientific) following the manufacturer's instructions. Equal concentrations of each library were sequenced using an Ion Proton System (Thermo Fisher Scientific).

**WES data pre-processing**. For all 50 donors, DNAs from tumor tissues and matched PBMCs were sequenced. The sequenced reads were mapped to the hg19 reference genome using BWA mem[42], and post-mapping procedures, including sorting and marking of duplicate reads, was performed using the Picard (http://broadinstitute.github.io/picard/). The average coverage was 244 ×, the average mapping rate was 99.63%, and the average on-target rate was 69.78%. Supplementary Data 1 presents the coverage, mapping rate, and reads-on-target rate for each sample. BAM-formatted files, including read sequences, sequence qualities, and alignment status, were generated by reference genome mapping using BWA mem[42] and the Picard. We developed and applied an in-house pipeline for the processing of raw sequences through the stages of mapping quality control and mutation calling. The utilized software included Mutect[43], Indelocator (https://www.broadinstitute.org/cancer/cga/indelocator), and Oncotator[44]. The results were summarized and formatted to mutation annotation format (MAF) specifications (https://wiki.nci.nih.gov/display/TCGA/Mutation+Annotation+Format+(MAF)+Specification). In total, 50 sets of tumor and matched normal samples were analyzed; from them, a total of 24,051 somatic mutations and 9883 reported germline single nucleotide variants (previously deposited in dbSNP[45]) were identified. Supplementary Data 3 lists all the somatic mutations found.

**Identification of putative driver genes**. The MutSigCV algorithm[46] was used to identify significant mutated genes based on the presence of mutations/silent mutations, the nucleotide context, gene expression, gene replication time, and mutations in surrounding regions. The background mutation rate was considered for each mutated gene. The driver genes were sorted by q-value, which was generated as the Benjamini–Hochberg false discovery rate. All parameters were set to default values, and the q-value was set to < 0.1. Putative driver genes identified based on cohort of OSCC-Taiwan. Supplementary Table 2 lists the q-values for the putative driver genes and percentage of patients having mutations in these genes in OSCC-Taiwan, OSCC-TCGA or OSCC-India.

**Validation of somatic mutations**. To validate the somatic mutations detected by our exome sequencing, we used the Ion PGM System (Life Technology) to investigate mutations located in 409 genes included in the Ion AmpliSeq Comprehensive Cancer Panel (CCP, Life Technology). The average sequencing depth was >1000 ×. The Ion Reporter software (variant caller V5.0.3.5; Life Technology) was used to perform variant calling and identify mutations across 49 matched tumor and normal samples. The CCP was found to contain 816 somatic SNPs. Details for these mutations is presented in Supplementary Data 4. For validation, we used 49 matched tumor and normal sample pairs and a total of 428 mutation sites covered by the CCP target-sequencing regions.

**Identification of copy CNVs**. After the sequence reads were mapped to hg19 with BWA[42], de-duplicated BAM files were generated from the original bam files using Picard. GATK DepthOfCOverage[47] was used to estimate coverage from the de-duplicated bam files. The coverage information from paired samples was then analyzed by exome CNV[48], which utilizes a circular binary segmentation algorithm[49] to calculate the mean coverage log ratio for each segment. All such ratios not belonging to a sex chromosome were then analyzed with GISTIC 2.0[50], which identifies statically significant amplified or deleted regions. Genes located within the putative amplified/deleted regions were further checked whether they are reported to be a driver mutation gene in HNSC in driverDB[51]. CNVs for 50 OSCC patients from India were obtained from a supplementary table of the relevant study[12].

**Identification of DEGs in OSCC samples from OSCC-Taiwan**. In first cohort, we have carried out RNA-Seq in normal/tumor paired sample in 39 out of the 50 patients. All RNA reads were first trimmed by Trimmomatic[52] and mapped to hg19 with STAR[53]. The statistics obtained for the RNA-Seq reads are shown in Supplementary Data 2. The expression levels of genes in the 39 normal/tumor pairs were estimated by RSEM[54, 55] with GENCODE annotation[56]. To accurately estimate the expression levels of *A3A*, *A3B*, and *A3A_B*, the genotype of each sample was taken into consideration for quantification. Genes having median transcripts per million (TPM) larger than 0.5 were used in DEG detection. A total of 3548 DEGs were selected with criteria adjusted p-value < 0.05 and fold change > 2 with Partek Genomics Suite software (Inc. P. Partek Genomics Suite. St. Louis). The 3548 identified DEGs were further used for hierarchical clustering analysis.

**Pathway analysis of the identified DEGs**. The 3548 DEGs were subjected to pathway enrichment analysis using MetaCore from Thomson Reuters, which uses hypergeometric testing to examine whether genes from various pathways are found more frequently than expected in a list of DEGs.

**Luciferase reporter assay**. *A3A* 3′UTR and *A3B* 3′UTR were PCR amplified from genomic DNA of OC3 oral cancer cell line, using the same forward primer 5′-ACTAGTAGGATGGGCCTCAGT CTCTAAG-3′; reverse primer 5′-AAGCTT AGTGTTTGTGGAAACTCTTGCAATT C-3′ is for *A3A* 3′UTR, and 5′-AAGCTT AGTGTTTGTGGAAACAATTATGGAAG-3′ is for *A3B* 3′UTR. The amplified products were cloned into the pMIR-Report-Vector (Ambion). Precursor of miR-409 was amplified from OC3 cell genomic DNA, and cloned into pcDNA6.2-GW/EmGFP-miR (Invitrogen). 293T cells ($2 \times 10^5$) were subjected to transient calcium-phosphate-mediated transfection with the following: 10 ng of pMIR-Lusiferase-A3B 3′UTR; 1 μg of the vector control or expression vectors encoding miR-409 and 10 ng of pCMV-Renilla (Promega). Luciferase and renilla activities were measured using the Dual-Luciferase Reporter Assay System (Promega). The luciferase values were normalized to those of renilla, and the results are presented as the luciferase/renilla ratio.

**Comparison of DEGs in OSCC-Taiwan and OSCC-TCGA**. To examine whether the identified DEGs exhibited similar expression changes in the TCGA transcriptome database (http://cancergenome.nih.gov/), we downloaded Level 3 RNASeqV2 data for HNSC. This dataset contains 315 OSCC samples that were taken from anatomic sites within the oral cavity and designated as squamous cell carcinoma. Among them, 30 samples also had transcriptomic data available from their normal tissue sample. We used the Partek Genomics Suite software to compare the expression levels of genes sharing the same Entrez IDs in all (315 tumor + 30 normal) samples.

**Mutational signature analysis**. The Wellcome Trust Sanger Institute (WTSI) Mutational Signature Framework[14] (MATLAB R2008b) and several in-house-developed R scripts were used to perform the mutational signature analysis. In addition to the 50 Taiwanese OSCC cases, we also analyzed data from 32 cancer projects deposited to the TCGA/ICGC data portal. To compare the mutational signatures across OSCC tumors, we extracted 172 TCGA-HNSC patients whose tumors were of the oral cavity according to an available supporting table[11], and used their data to generated a data subset consisting of 26,050 somatic SNV mutations (designated OSCC-TCGA). The signature stability cutoff used to determine the number of mutational signature profiles for each cancer project was

set to 0.85. The non-negative matrix factorization calculation was performed using the WTSI framework with default parameters. To identify the representative signature catalogs, a cosine similarity algorithm was used to compare the predicted signature spectra with the 27 mutational signatures previously defined by Alexandrov's group[13, 57]. According to the biological features reportedly associated with mutational signatures[8, 11] we distributed all of the signatures into four biological categories: the APOBEC-associated signature (signatures 2 and 13), the smoking-related signature (signatures 4 and 5), the aging-associated signature (signatures 1A and 1B), and others (all other representative signatures). The contributions of each functional category were calculated by summing the individual contributions of each mutational signature in a given category.

**RNA-Seq-based detection of *A3A_B* fusion transcripts**. To detect *A3A_B* fusion transcripts, we adapted a previously described RNA-Seq read mapping approach[58]. Specifically, we used the BWA mem program of BWA 0.7.12[42] to map RNA-Seq reads to a database that integrated the hg19 reference genome and all annotated putative splicing junctions. Specifically, exonic sequences encompassing known splicing junctions derived from annotations of GENCODE V19[56], RefSeq, Ensembl archive 75, and UCSC hg19 KnownGene were combined into index database to avert concurrent hits to exonic sequences that containing splicing junction regions and the reference genome. For the 151-bp RNA-Seq reads studied in the present work, an upstream 150 bp and another downstream 150 bp (1 bp shorter than the length of RNA-Seq reads) of exonic sequences surrounding a known splicing junction were extracted. While upstream and/or downstream neighboring exons of a known splicing junction were shorter than the required length (150 bp in this study), we included more exons to extend the regions to desired length. For each mapped bam file, only uniquely mapped reads were considered. We then used MarkDuplicates of Picard 1.136 to remove duplicate reads that mapped to the same location; in these cases, we retained the read with the best mapping quality for further analysis. To display reads that spanned the splicing junctions of *A3A*, *A3B*, and *A3A_B*, we used the Sashimi plot of Integrated Genome Viewer 2.3.77[59] to visualize the spliced reads and the *A3A_B* fusion form.

**Statistical analysis**. Patient characteristics were stratified using various clinicopathological factors and evaluated by the chi-square test or Wilcoxon test. Multivariate analyses were applied to define OS, DSS and DFS. Survival rates were estimated by Kaplan–Meier plotting and compared by log-rank test. Statistical analyses were performed using SAS software (v.9.3). All patients received follow-up consultations at our outpatient clinic until March 2016 or death. All $p$-values were two-sided, and the significance level was set at $p < 0.05$.

**Protein quantification**. Each tissue specimen was homogenized with 0.1% RapiGest buffer (Waters, Bedford, MA, USA) using a bead-beating homogenizer (Precellys 24; Bertin Technologies, Ozyme, France). The tissue extracts were then digested with trypsin (Promega, Madison, WI, USA) and labeled with the iTRAQ reagent according to the manufacturer's protocol. The iTRAQ 114 reagent was used to label the digestion product of 30-pairs of OSCC tissues, and iTRAQ 115 and 116 reagents were used to label peptides of non-tumor tissue and tumor tissue, respectively. The labeling mixtures were desalted with an in-house-built C18-microcolumn and analyzed using on-line 2D-HPLC (Dionex Ultimate 3000; Thermo Fisher) coupled to a 2D linear ion trap mass spectrometer (LTQ-Orbitrap ELITE; Thermo Fisher)[60]. The data analysis was carried out using Proteome Discoverer software with the node of reporter ion quantifier for iTRAQ quantification (version 1.4, Thermo Fisher Scientific). The MS/MS spectra were searched against the GENECODE V19 human sequence database using the Mascot search engine (version 2.5; Matrix Science, London, UK). Peptide-spectrum matches were filtered to ensure that the overall false-discovery was < 0.01. Proteins identified as A3A with a single peptide hit were removed. The quantitative data for A3A protein were exported from Proteome Discoverer via integration of the most confident centroid with a 20-ppm tolerance. The A3A protein levels in each pair of tumor/non-tumor tissue pair (116/115) were taken as the median ratio of identified A3A peptides, and subjected to global normalization.

**CNV analysis for OSCC-TCGA**. Somatic CNVs for HNSC samples were available in TCGA database. As this project focused on OSCC, we selected 315 samples taken from the alveolar ridge, floor of mouth, buccal mucosa, hard palate, lip, tongue, or oral cavity. To assess the selected samples for CNVs, the segment means derived using Affymetrix Genome-Wide SNP Array 6.0 were downloaded from TCGA (http://cancergenome.nih.gov/) and analyzed with GISTIC 2.0[50], and the genes located within putative amplified/deleted regions were checked whether they are reported to be a driver mutation gene in HNSC in driverDB[51].

**RNA extraction, quantification, and fusion gene detection**. Paired OSCC tumor/pericancerous normal tissues were individually homogenized in liquid nitrogen and subjected to total RNA extraction using RNAzol B (Tel-Test, INC.), according to the manufacturer's protocol. First-strand complementary DNA (cDNA) was synthesized from 5 µg of total RNA using oligo dT primers and SuperScript III RT and qRT-PCR analysis was carried using the QuantStudio 12K Flex Real-time PCR System (Applied Biosystems). In 10 µl of reaction volumes containing 2 µl of 10-fold diluted cDNA, 5 µl of 2 × master mix (Roche), and 0.15 µM of primers. The utilized primers included: A3A-q2F (5′-ATGGCATTG GAAGGCATAAG-3′) and A3A-q2R (5′-CAAAGAAGGAACCAGGTCCA-3′) for detection of *A3A*; A3B-qrtF (5′-GACCCTTTGGTCCTTCGAC-3′) and A3B-qrtR (5′-GCACAGCCCCAGGAGAAG-3′) for detection of *A3B*; and TBP-F849 (5′-TGCTCACCCCACCAACAATTTAG-3′) and TBP-R969 (5′-CTGGGTTTGA TCATTCTGTAGATTAA-3′) for detection of *TPB* as an internal control. The qRT-PCR conditions were as follows: 95 °C 10 min, followed by 40 cycles of 95 °C for 30 s, 60 °C for 1 min.

*A3A_B* fusion gene was amplified from cDNA by PCR using primers A3A_B5′ (5′-ATGGAAGCCAGCCACC-3′) and A3A_B3′ (5′-TCAAATTAAAATTAAT TGACTCTGATT-3′) and the PCR conditions: 95 °C for 10 min, followed by 35 cycles of 95 °C for 30 s, 60 °C for 30 s, and 68 °C for 30 s, and a final soak at 72 °C for 10 min. The 800-bp PCR product was examined by gel electrophoresis, followed by validation with Sanger sequencing.

**Genotyping of the *APOBEC3B*-deletion polymorphism**. For OSCC-Taiwan cohort, genomic DNA was obtained from PBMCs of OSCC patients, and 20 µg of each sample was used for PCR-based *A3A* genotyping. Primers Del F (5′-TAG GTGCCACCCCGAT-3′) and Del 2R (5′-CTAAATATGGAGCCAATTAA-3′) were used to generate a 757-bp product as an indicator of the deletion poly-morphism, while primers Ins 2F (5′-TGTCCCTTTTCAGAGTTTGAGTA-3′) and Ins 2R (5′-TCCTTAGAGACTGAGGCCCAT-3′) were used to generate a 449-bp product that represented the wild-type chromosome. The PCR conditions were as follows: 95 °C for 10 min, followed by 37 cycles of 95 °C for 15 s, 55 °C (for the deletion polymorphism) or 60 °C (for the wild-type sequence) for 30 s, and 72 °C 30 s, followed by a final soak at 72 °C for 10 min. The obtained products were analyzed by agarose gel electrophoresis.

For all the HNSC samples having WES data available in TCGA (phs000178.v9. p8) include 318 unique samples taken from anatomic sites within the oral cavity and designated as representing OSCC. We download controlled-access WES bam files for the OSCC samples of these 318 cases from the GDC websites (https://gdc. cancer.gov/, Data Release V2). To identify *APOBEC3B*-deletion polymorphism from the bam files, we used the expectation-maximization approach to model the copy number of each sample based on the ratio of the sequencing depths inside and outside the deletion region[22].

**Data availability**. The sequencing data have been deposited in the NCBI Sequence Read Archive (SRA) database under the accession code SRP078156. The data of WES, RNASeqV2 and Affymetrix Genome-Wide SNP Array 6.0 referenced during the study are available in a public repository from the TCGA (http://cancergenome. nih.gov/) website. The authors declare that all the other data supporting the findings of this study are available within the article and its supplementary information files and from the corresponding author upon reasonable request.

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

## Acknowledgements

This study was supported by: grants (MOST102-2628-B-182A-012-MY3, MOST103-2632-B-182-001, MOST104-2632-B-182-001, MOST105-2632-B-182-001, MOST104-2321-B-182-007-MY3 and MOST104-2320-B-182-033) from the Ministry of Science and Technology; grants (EMRPD1G0031 and 0141) from the Ministry of Education; and grants (CMRPG3F0152, CMRPG3C1913, CIRPG3B0012, CIRPG3B0013, CIRPD3B0012, CMRPD1D0101, CMRPD1D0102, CMRPD1D0103, CMRPD3E0071, CMRPD3E0072, CMRPG3D1511, CMRPG3D1512, and CMRPG3D1513) from Chang Gung Memorial Hospital, Taiwan. We thank all of the members of the Cancer Center at Chang Gung Memorial Hospital and the Pathology Core of the Chang Gung Molecular Medicine Research Center for their invaluable help. We would also like to thank National Center for High-Performance Computing of the National Applied Research Laboratories of Taiwan and the National Research Program for Biopharmaceuticals (MOST 104-2325-B-492-001) for providing computational biology platform. The results published here are partly based on data generated by TCGA managed by the NCI and NHGRI. Information about TCGA and GDC can be found at http://cancergenome.nih.gov and http://gdc.nci.nih.gov/, respectively. We thank Dr. David Wedge from the Wellcome Trust Sanger Institute, Wellcome Trust Genome Campus, Hinxton, UK for providing the EM algorithm.

## Author contributions

H.L., K.-P.C., and Y.-S.C. designed the study. H.L., C.-S.W., C.-D.C., H.-P.L., T.E.H.C., C.-Y.Y., and C.-W.H. performed the experiments. T.-W.C., C.-C.L., C.-R.P., P.-J.H., J.W., I.Y.-F.C., Y.-M.Y., L.-J.C., B.Z., W.-C.L., C.-C.W., Y.-T.C., J.-D.L., C.H., and J.-S.Y. analyzed the data. T.-W.C., L.-C.S., and K.-P.C. performed statistical analyses.

J.W., B.Z., and P.T. contributed analytic tools. T.-W.C., C.-C.L., H.L., C.-S.W., B.C.-M.T., K.-P.C., and Y.-S.C. wrote the paper, with contributions from W.-F.C., H.R., and J.-N.M.

## Additional information

**Competing interests:** The authors declare no competing financial interests.

Ting-Wen Chen[1,2], Chi-Ching Lee[1,2,3], Hsuan Liu[1,4,5,6], Chi-Sheng Wu[1,7], Curtis R. Pickering[8], Po-Jung Huang[1,2,9,10], Jing Wang[11], Ian Yi-Feng Chang[1,2], Yuan-Ming Yeh[1,2], Chih-De Chen[1], Hsin-Pai Li[1,4,12,13], Ji-Dung Luo[1,2], Bertrand Chin-Ming Tan[1,4,9,14], Timothy En Haw Chan[4], Chuen Hsueh[15,16], Lichieh Julie Chu[1,17], Yi-Ting Chen[1,4,9], Bing Zhang[18], Chia-Yu Yang[1,6,12], Chih-Ching Wu[1,7,19], Chia-Wei Hsu[1], Lai-Chu See[20,21], Petrus Tang[1,2,22,23], Jau-Song Yu[1,17,24], Wei-Chao Liao[1,7], Wei-Fan Chiang[25,26], Henry Rodriguez[27], Jeffrey N. Myers[7], Kai-Ping Chang[7,28] & Yu-Sun Chang[1,4,7]

[1]Molecular Medicine Research Center, Chang Gung University, Guishan, Taoyuan 33302, Taiwan. [2]Bioinformatics Center, Chang Gung University, Guishan, Taoyuan 33302, Taiwan. [3]Department and Graduate Institute of Computer Science and Information Engineering, Chang Gung University, Guishan, Taoyuan 33302, Taiwan. [4]Graduate Institute of Biomedical Sciences, Chang Gung University, Guishan, Taoyuan 33302, Taiwan. [5]Department of Biochemistry, Chang Gung University, Guishan, Taoyuan 33302, Taiwan. [6]Division of Colon and Rectal Surgery, Chang Gung Memorial Hospital, Linkou, Gueishan, Taoyuan 33305, Taiwan. [7]Department of Otolaryngology-Head & Neck Surgery, Chang Gung Memorial Hospital at Linkou, Gueishan, Taoyuan 33305, Taiwan. [8]Departments of Head and Neck Surgery, the University of Texas MD Anderson Cancer Center, Houston, Texas 77030, USA. [9]Department of Biomedical Sciences, Chang Gung University, Guishan, Taoyuan 33302, Taiwan. [10]Genomic Medicine Core Laboratory, Chang Gung Memorial Hospital at Linkou, Gueishan, Taoyuan 33305, Taiwan. [11]Departments of Biostatistics, the University of Texas MD Anderson Cancer Center, Houston, Texas 77030, USA. [12]Department of Microbiology and Immunology, Chang Gung University, Guishan, Taoyuan 33302, Taiwan. [13]Division of Hematology-Oncology, Chang Gung Memorial Hospital at Linkou, Gueishan, Taoyuan 33305, Taiwan. [14]Department of Neurosurgery, Chang Gung Memorial Hospital at Linkou, Gueishan, Taoyuan 33305, Taiwan. [15]Pathology Core of the Molecular Medicine Research Center, Chang Gung University, Guishan, Taoyuan 33302, Taiwan. [16]Department of Pathology, Chang Gung Memorial Hospital at Linkou, Gueishan, Taoyuan 33305, Taiwan. [17]Liver Research Center, Chang Gung Memorial Hospital at Linkou, Gueishan, Taoyuan 33305, Taiwan. [18]Department of Molecular and Human Genetics Lester & Sue Smith Breast Center, Baylor College of Medicine, Houston, Texas 77030, USA. [19]Department of Medical Biotechnology and Laboratory Science, Chang Gung University, Guishan, Taoyuan 33302, Taiwan. [20]Department of Public Health, Chang Gung University, Guishan, Taoyuan 33302, Taiwan. [21]Biostatistics Core Laboratory, Chang Gung University, Guishan, Taoyuan 33302, Taiwan. [22]Molecular Regulation and Bioinformatics Laboratory, Chang Gung University, Guishan, Taoyuan 33302, Taiwan. [23]Molecular Infectious Diseases Research Center, Chang Gung Memorial Hospital at Linkou, Gueishan, Taoyuan 33305, Taiwan. [24]Department of Cell and Molecular Biology, Chang Gung University, Guishan, Taoyuan 33302, Taiwan. [25]Department of Oral & Maxillofacial Surgery, Chi-Mei Medical Center, Liouying 736, Taiwan. [26]School of Dentistry, National Yang Ming University, Taipei 112, Taiwan. [27]Office of Cancer Clinical Proteomics Research, National Cancer Institute, US National Institutes of Health, Bethesda, Maryland 20892, USA. [28]College of Medicine, Chang Gung University, Guishan, Taoyuan 33302, Taiwan. Ting-Wen Chen, Chi-Ching Lee, Hsuan Liu, Chi-Sheng Wu, Curtis R. Pickering, Kai-Ping Chang and Yu-Sun Chang contributed equally to this work

