## [Peer Review File · Nature Communications]

Reviewers' comments:

Reviewer #1 (Remarks to the Author):

The manuscript by Chen et al. is describing the tumor characteristics of a Taiwanese OSCC cohort that is enriched for the APOBEC3B germline deletion through various NGS strategies. The authors also compare this cohort with previously published TCGA and Indian data to identify unique characteristics of their cohort. They purport to find that there is an association with the increased expression of A3A and the A3A_B fusion transcript with the prevalence of APOBEC signature mutations in their cohort. The increased expression of A3A also appears to correlate with better outcome and specific clinicopathological characteristics of the tumors in HE and HO patient groups. Overall, this is a very interesting cohort, but there are several major reservations that may undermine and/or change the overall conclusions of the study.

Major comments

1) Fig 1a should show all 6 types of base substitution mutations for clarity. At this time the data in Fig 1a and Fig 2 do not appear to be concordant as major groups of mutations are excluded for unknown reasons.

2) Results in Fig 4f and corresponding primer designs are questionable. The relative values (>100) are very high compared to previously published and validated assays. Since the assays for A3A and A3B expression, in particular, appear to be a novel, the appropriate experiments validating their specificity and efficiency need to be included especially considering the high degree of homology between APOBEC3 family members.

3) The authors should definitely check if there are other germline variants in the APOBEC locus that may contribute to some of the unique characteristics of the OSCC-Taiwan cohort compared to the TCGA and Indian cohorts. Most importantly, please check to see if novel variants are in linkage disequilibrium with the deletion polymorphism as is quite possible in an ethnically similar population. Other germline variants have been identified recently by Middlebrooks et al. *Nat. Gen.* 2016 and Starrett et al. *Nat. Comm.* 2016 that explain APOBEC signature mutations. Minimally these variants and maximally all variants within the entire APOBEC3 locus should be compared.

4) The authors should evaluate and control for the infiltration of immune cells and stroma, which have been shown to affect the expression of APOBEC-family genes and correspond with clinical outcome (Leonard et al. *CCR.* 2016 and Smid et al. *Nat. Comm.* 2016). This should be done using both RNAseq and IHC to accurately assign expression of APOBEC family members to tumor/immune/stromal cells. In particular, IHC experiments using tissues from this unique cohort will unambiguously determine which cells are responsible for the abnormally high A3A signal (ie. is this the actual tumor cells or infiltrating macrophage lineage cells). Several commercial academic antibodies are available for these purposes.

Minor comments

5) HO/HE/non nomenclature is confusing. A3B-/-, A3B+/-, A3B+/+ would be much clearer.

6) Subpanel designation (a, b, c, etc.) should be larger to more easily interpret figures.

7) Line 59. Should include a reference for how betel nut chewing is a risk factor.

8) Line 179. What about APOBEC3B peptides? Supplementary Fig 8. The peptide in the bottom row is identical in A3A, A3B, and A3G via protein BLAST. The peptide in the middle row is only 1 AA different from that seen most A3G entries in the nr protein database and is identical to the AA sequence used in an A3G structure (PMID: 25542899). Also an example of the raw data for the control should be shown.

9) Line 189. APOBEC3A.

10) Line 194. Authors should include these data. Also change wording to indicate that miRs destabilize mRNA transcripts rather than their absence stabilizes mRNA transcripts.

11) Line 203. Proper statistical test should be applied to see if YTCW:RTCW ratio is significantly different. Also the cited paper used YTCA:RTCA ratio as an enrichment metric for APOBEC3A-mediated mutagenesis. Lastly these data should be subdivided by HO/HE groups.

12) Line 229. This line makes it sound as if A3B transcript is being detected. It should be clarified that this detection is due to mismapping of A3A reads. Considering this high rate of cross mapping I would also like to see additional validation of unique A3B and A3A expression in this cohort. These reads and transcript levels should be excluded from analyses and graphs since they are clearly artifactual.

13) Line 326. The reference is referring to a fusion construct containing the the coding region of A3A and the A3B 3'UTR, not just the UTR.

14) Line 332. Logically this miRNA should also suppress the A3A transcript due to homology in the UTR.

15) Lines 331 & 334 Why are the data not shown?

16) Fig 3d should be removed from main figure set. It does not add valuable information to the paper, nor is adequately addressed for a mechanistic explanation for A3A contributing to cancer mutagenesis in this manuscript.

17) Fig 4a-d. This is a lot of validation and can be more concisely summarized and mostly moved to a supplemental figure.

18) Why in Fig 5 do the A3A low expressers in HE and HO have drastic survival differences. Wouldn't it be expected if this was due to A3A that low expression would be nearly identical?

19) Supplementary methods p21. Both primers for A3B are referred to as A3B-qrtF.

20) Supplemental fig S9 should also show correlation coefficient as well as highlight all samples in this study by A3B deletion status and any other impactful mutations (such as those in p53).

21) Samples with <200X coverage and ~40% reads mapped to target are questionable for such a consistently high coverage HiSeq run. Do these look like outliers in the data analysis?

22) The authors should also consider subsetting TCGA samples by cancer stage to better compare to their cohort, which is >60% late-stage cancer (4A).

Reviewer #2 (Remarks to the Author):

Chang et al describe a germline variant of the APOBEC genes A3A and A3B in the Taiwanese population, and show that this seems to have consequences for the mutational profile in oral cancers. The authors did a lot of work. The findings are interesting but some critical information is still missing.

Comments:

1) Is Figure 1 not mislabeled with respect to deletion and amplification (copy number)? Usually CDKN2A is deleted and EGFR amplified. Please correct and check labels everywhere. Should the Apobec cases not be identified in the same Fig?

There are 4 HPV cases identified, which all have a TP53 mutation. This is highly unlikely. How was HPV status defined, only by NESTED E6 RT-PCR. Very unreliable. E6 RT-PCR on RNA should be used. Alternatively, the authors have RNAseq data and exome data and should map the reads against HPV. My guess is that they are all negative. If they are truly positive these cases should be marked in the subsequent analyses as these are typically Apobec-type tumors.

The authors used the Lawrence filter on their data in Sup Fig 6, but present unfiltered mutations already in Sup Fig2. I noted TTN as frequently mutated gene which is a typical suspect. They should warn in Sup Fig 2 that this is unfiltered data. Which VAF was used to call a mutation?

2) The authors define that 38% of the cases showed an Apobec signature, but this is critical information and should be displayed in Figure 2 per case, including how it was defined. Just an algorithm score? How did these scores vary, also in respect with HO or HE? See also point 4.

3) Sentence 168 and 169 are very confusing. The authors state that Apobec is typically OSCC while it is absent in the Indian OSCC cohort?

4) I miss a Figure that associates the Apobec score of the tumor against the genotype HO, HE or non. There are only associations with A3A and A3B expression while the key point is the association of genotype (and HPV status) with Apobec score.

5) The Apobec genotype is in the germline and not somatic. Hence a difference in expression between tumor and normal is not to be expected. The authors suggest that this is related with the A3B 3'UTR and aberrant microRNA expression in the tumor. An interesting hypothesis but it should be proven in a functional experiment and shown that the fusion gene is indeed expressed at a higher level in tumor cells than in normal cells. Obviously this requires some reference transfection by a marker gene, but it is do-able as mucosal keratinocytes can be cultured and transfected.

6) The clinical association study should be described in much more detail. How were patients treated? Were all margins histologically tumor-negative? How were recurrences and particularly second primary tumors evaluated? How frequent was the followup?

Reviewer #3 (Remarks to the Author):

Chen et al explore the specific relationship between a common deletion allele which results in the fusion of APOBECA and B genes (deletion of A3B) in the Taiwanese population with oral cancer. This is a particularly interesting cancer to study given the unique environmental factors that influence OSCC (betel nut chewing) as well as the increasing knowledge of mutation patterns and is clinically very significant given the high morbidity and mortality. The authors highlight that a frequent germline polymorphism which results in fusion of APOBEC3A and the 3'UTR of APOBEC3B is frequent in some populations (including the Taiwanese) and may influence the biology of the tumor development. The authors also provide some initial data that presence of this polymorphism (particularly when homozygous) may also influence survival. This study highlights the increasing complexity of understanding cancer genomics as these studies extend from the initial cohorts of European whites to many different global populations.

Major

1. The abstract is very hard to follow if someone isn't familiar with the gene cluster. It would be good to clarify the source of the fusion transcripts from the deletion as well as the patient population being studied (and how many have deletion alleles). The abstract also doesn't convey that there are two different independent sets being studied.

2. For any paper describing both germline and somatic mutations it is very helpful to be very clear when you are referring to each type of data. For example, is the following sentence referring to unique somatic mutation pattern or germline variation from the local population. It appears from the figure legend to be somatic but it would be helpful to clarify in the text.

“Interestingly, however, our analysis further identified mutations in seven genes (LCE4A, ORAI1, ZNF717, PABPC3, SKA3, ODF1, and SETD8) that turned out to be unique, locally prevalent ($\geq 10\%$ of patients) ”

3. It is an important point that the gene expression of A3B detected in some studies results from transcripts from the 3'UTR. A comment in the Discussion to researchers doing expression analysis to not assume that a gene is over-expressed without being clear that the coding region itself is being over-expressed would be useful.

4. In the second set of tumors, the authors state that all of the tumors had over-expression of A3A or A3B. However, the mutation pattern associated with APOBEC was only seen in 38% of the first cohort. Given the frequency of the A3B deletion allele in the larger set why is it only a minority that have this mutation spectrum? Similarly, the authors never comment on whether these alleles are enriched in this tumor population versus the ethnic Taiwanese population as a whole. This should be clearly stated (or the reason it isn't studied provided).

5. For Figure 4 panel E – do the authors want to discuss why they think there is such a wide range of expression of both A3A and A3B in the tumor samples.? Since this is stated to be done by RT-PCR, why couldn't the authors use a PCR that would not detect the deletion allele?

6. Why do you think that being heterozygous for the deletion allele isn't enough to give increased expression of A3A? Given that there are three genotypes why did you decide to use just two categories (high and low) for the survival analysis. Did the expression cut-off used mirror what you see with the homozygous deletion allele carriers or include the heterozygous carriers?

7. The survival data is being assessed from a relatively small cohort and it isn't clear if multi-testing was accounted for in all the different analyses being done. The authors may want to make a statement that the survival data would need to be replicated in a larger cohort.

Minor:

1. Which “Americans” are being referred to in the abstract. Do you mean Amerindians? The white US population typically resembles European in frequency.

2. For the international audience of Nature Communications it would be helpful to describe the origins of “ethnic Taiwanese”

3. You may want to comment in the Discussion that almost all of the patients also have a history of cigarette smoking in addition to betel nut chewing.

REVIEWERS' COMMENTS:

Reviewer #1 (Remarks to the Author):

The manuscript by Chen et al. is describing the tumor characteristics of a Taiwanese OSCC cohort that is enriched for the APOBEC3B germline deletion through various NGS strategies. The authors also compare this cohort with previously published TCGA and Indian data to identify unique characteristics of their cohort. They purport to find that there is an association with the increased expression of A3A and the A3A_B fusion transcript with the prevalence of APOBEC signature mutations in their cohort. The increased expression of A3A also appears to correlate with better outcome and specific clinicopathological characteristics of the tumors in HE and HO patient groups. Overall, this is a very interesting cohort, but there are several major reservations that may undermine and/or change the overall conclusions of the study.

Major comments

1) Fig 1a should show all 6 types of base substitution mutations for clarity. At this time the data in Fig 1a and Fig 2 do not appear to be concordant as major groups of mutations are excluded for unknown reasons.

Response:

Thanks for the reviewer's suggestion. We modified the top panel of Fig. 1a by adding all 6 types of base substitutions, which are color-coded. We also revised the Fig. 1b, and present the significantly mutated genes sorted by their q values in 50 Taiwan-OSCC samples. The modification was made due to the following reasons. In the original analyses, we combined the results of InDels obtained from two analytical tools, VarScan2 and Indelocator. In this case, VarScan2 can identify much more InDels than the Indelocator (9,109 vs. 2,748). However, more than 40% InDels identified by VarScan2 show less than 10% differences between tumor and normal allelic fractions (Response Fig. 1) in the 50 matched samples examined in this study. Moreover, VarScan has not recommended for InDels identification according to

the NIH GDC documentation

(https://docs.gdc.cancer.gov/Data/Bioinformatics_Pipelines/DNA_Seq_Variant_Calling_Pipeline/). Therefore, we removed those InDels only identified by VarScan2. The results did not alter the list of the most significantly mutated genes, which are *TP53*, *FAT1*, *NOTCH1*, *PIK3CA*, *CDNK2A*, *DHRS4*, *RASA1*, *SETD8*, *HRAS* and *CENPV* as calculated by MutSigCV. In addition, *HRAS* was picked by the MutSigCV among the top 10. The mutations in all 10 genes were confirmed by CCP, Sanger sequencing or pyrosequencing. More importantly, removal of the InDels identified by VarScan2 didn't change the results of the mutational signatures; only SNVs were considered in mutational signature calculations. Accordingly, we made modifications in the text, on-line methods, Supplementary Fig. 2a and 3, Supplementary Table 4 and 6.

Response Figure 1. The percentage of difference in allelic fractions (ΔF , Tumor allelic fractions - Normal allelic fractions of each somatic mutation in paired samples). There are 9,109 InDels identified by VarScan2 and 2,748 InDels identified by Indelocator. There are only 340 InDels identified by both VarScan2 and Indelocator. In this summary plot, InDels were shown by either VarScan2 or Indelocator. The X-axis represents the ΔF , 10% as interval. The percentage for InDels identified by Indelocator and VarScan2 are shown in blue and orange, respectively.

The revised fig. 1 is as follows.

Fig. 1 in the revised manuscript. At the bottom of the revised Fig. 1, we also added the genotyping results of each patient recruited to this study (as suggested by the Reviewer #2).

The comparison of the mutations in 10 genes from the original Figure 1 but not present in the revised Figure 1 are shown in the Response Figure 2.

Response Figure 2. Mutation profiles for genes with and without the InDels identified by VarScan2. V1 (version 1) shows mutation and InDels predicted with Mutect, Indelocator and VarScan2 (in Fig.1 of the original manuscript). V2 (version 2) shows mutation and InDels predicted with Mutect and Indelocator (in the Figure 1 of the revised manuscript). Mutation rates are present at the end of each row. Red and blue boxes represent mutations with nonsense/frame-shift/splice-site annotation and missense/in-frame InDels respectively. Mutations that were identified in V1 but absent in the revised manuscript are present in light red and light blue.

2) Results in Fig 4f and corresponding primer designs are questionable. The relative values (>100) are very high compared to previously published and validated assays. Since the assays for A3A and A3B expression, in particular, appear to be a novel, the appropriate experiments validating their specificity and efficiency need to be included especially considering the high degree of homology between APOBEC3 family members.

Response:

We appreciate the reviewer's comments. In Figure 4c (the original Fig. 4f), the primers used to amplify the transcripts of *A3A* and *A3B* by RT-qPCR have been confirmed with their locations on chromosome 22, and the PCR products with unique sequence specific to *A3A* and *A3B* genes have been confirmed by Sanger sequencing. Briefly, for *A3A*, the 176 bp RT-PCR products covering the sequences from exon 3 to exon 4 of *A3A* gene were generated using qPCR primers A3A_q2F and A3A_q2R, which located on chr. 22(+): 39,355,588-39,355,607 and chr.22(-):39,357,444-39,357,463, respectively (Response Fig. 3a). For *A3B*, the 256 bp RT-PCR products crossed from exon 5 to exon 6 of *A3B* gene was generated using qPCR primers A3B_qRTF and A3A_qRTR, which located on chr. 22(+): 39,385,505-39,385,523 and chr. 22(-):39,387,461 39,387,482, respectively (Response Fig. 3b).

Response Figure 3. Sanger sequencing results of (a) A3A and (b) A3B qRT-PCR examined in this study. Both the A3A and A3B shown here were originated from qRT-PCR product from Taiwan-OSCC clinical samples.

3) The authors should definitely check if there are other germline variants in the APOBEC locus that may contribute to some of the unique characteristics of the OSCC-Taiwan cohort compared to the TCGA and Indian cohorts. Most importantly, please check to see if novel variants are in linkage disequilibrium with the deletion polymorphism as is quite possible in an ethnically similar population. Other germline variants have been identified recently by Middlebrooks et al. *Nat. Gen.* 2016 and Starrett et al. *Nat. Comm.* 2016 that explain APOBEC signature mutations. Minimally these variants and maximally all variants within the entire APOBEC3 locus should be compared.

Response:

Thanks for the reviewer's suggestion. We examined the germline variants (SNPs) in the APOBEC locus. There are 3,803 SNPs (dbSNP 135) located in APOBEC locus (from 20-kb up-stream and 20-kb down-stream of the APOBEC3 gene cluster). Among them, 516 SNPs were covered in the WES capture kit in this study. From these 516 SNP loci, 403 SNP loci were confidently detected with genotyping quality larger than 30 and depth larger than 10 in more than 45 patients from our 50 OSCC-

Taiwan cohort. Among these 403 SNPs, 50 SNPs are informative, or not having the same genotype in all patients. We then tested whether there are association between these 50 SNPs with (1) TCW/YTCW/RTCW patterns, (2) expression level of *A3A* and *A3B* expression level, (3) clinical indicators (stage/lesion/gender) and (4) *A3B* deletion genotypes with Kruskal–Wallis test (numerical data) or Chi-square test (categorical data). We only found one moderate association: *A3B* expression level is elevated in patients with rs139293 T/T ($p=0.049$, Kruskal–Wallis test). Notably, there are only 2 patients carrying this T/T at this position and this association is no longer significant after multiple test correction. In the future, it may be worthwhile to verify this trait with more cases.

The 6 SNPs reported by Starrett *et al.* (Nat. Comm. 2016) were all covered in our WES data. Among them rs34522862 was found to be T/T in all our 50 samples and the other 5 SNPs: rs139293, rs139297, rs139299, rs139298 and rs139302 are informative. As mentioned above, the rs139293 was found to be correlated with *A3B* expression level ($p=0.048$; Kruskal–Wallis test), but not correlated with other factors tested. All the other 4 SNPs were not associated with clinical indicators, APOBCE induced mutation (TCW/YTCW/RTCW patterns), expression level of *A3A* or *A3B* or *A3B* deletion polymorphism (Response Fig. 4, 5 and Response Table 1).

As for the three SNPs, rs17000526, rs1014971 and rs1004748, reported by Middlebrooks *et al.* (Nat. Gen. 2016), they are all located at the upstream of *A3A* and not covered in our WES. We then used Sanger sequencing to determine the genotypes. The results indicated that none of them are found to be associated with clinical indicators, APOBCE induced mutation (TCW/YTCW/RTCW patterns), expression level of *A3A* or *A3B* or *A3B* deletion polymorphism (Response Fig. 6, 7 and Response Table 2).

Response Figure 4. Association test between SNPs and APOBEC mutations.

There is no association between the 5 SNPs reported by Starrett *et al.* (Nat. Comm. 2016) and the APOBEC mutations. The p values derived from Kruskal–Wallis test were shown in each panel.

Response Figure 5. Association test between SNPs and expression levels of A3A and A3B. Among all the 6 SNPs reported by Starrett *et al.* (Nat. Comm. 2016), rs139293 was moderately correlated with A3B expression level with $p=0.049$ (Kruskal–Wallis test). No association were found between the other 4 SNPs and the expression level of A3A or A3B.

Response Table 1. Association test of the the 5 SNPs (reported in Starrett *et al.* Nat. Comm. 2016) and the clinical indicators/A3B deletion polymorphism.

Lesion	rs139302 (p=0.9689)			rs139293 (p=0.1143)			rs139297 (p=0.9167)			rs139298 (p=0.9167)			rs139299 (p=0.9167)		
	G/G	G/C	C/C	G/G	G/T	T/T	G/G	G/C	C/C	A/A	A/G	G/G	G/G	G/C	C/C
buccal mucossa	6	8	3	12	5	1	8	7	3	8	7	3	8	7	3
gingiva	1	2	2	4	0	0	1	2	2	1	2	2	1	2	2
gum	1	1	0	1	1	0	1	1	0	1	1	0	1	1	0
hard palate	1	0	1	1	1	0	1	0	1	1	0	1	1	0	1
mouth floor	0	1	0	0	0	0	0	1	0	0	1	0	0	1	0
retromolar trigone	1	1	1	2	1	0	1	1	1	1	1	1	1	1	1
tongue	5	8	3	13	4	0	8	8	3	8	8	3	8	8	3

Stage	rs139302 (p=0.6704)			rs139293 (p=0.8474)			rs139297 (p=0.5056)			rs139298 (p=0.5056)			rs139299 (p=0.5056)		
	G/G	G/C	C/C	G/G	G/T	T/T	G/G	G/C	C/C	A/A	A/G	G/G	G/G	G/C	C/C
I	0	2	0	2	0	0	0	2	0	0	2	0	0	2	0
II	3	3	2	6	1	0	3	3	2	3	3	2	3	3	2
III	2	1	1	2	1	0	2	1	1	2	1	1	2	1	1
IVA	7	13	7	19	9	2	11	13	7	11	13	7	11	13	7
IVB	3	2	0	4	1	0	4	1	0	4	1	0	4	1	0

Gender	rs139302 (p=0.457)			rs139293 (p=0.714)			rs139297 (p=0.426)			rs139298 (p=0.426)			rs139299 (p=0.426)		
	G/G	G/C	C/C	G/G	G/T	T/T	G/G	G/C	C/C	A/A	A/G	G/G	G/G	G/C	C/C
Male	13	18	10	28	11	2	17	17	10	17	17	10	17	17	10
Female	2	3	0	5	1	0	3	3	0	3	3	0	3	3	0

A3B	rs139302 (p=0.2282)			rs139293 (p=0.5196)			rs139297 (p=0.1952)			rs139298 (p=0.1952)			rs139299 (p=0.1952)		
	G/G	G/C	C/C	G/G	G/T	T/T	G/G	G/C	C/C	A/A	A/G	G/G	G/G	G/C	C/C
-/-	2	6	0	7	1	0	2	6	0	2	6	0	2	6	0
-/+	8	9	7	18	5	1	10	8	7	10	8	7	10	8	7
+/+	5	6	3	8	6	1	8	6	3	8	6	3	8	6	3

Response Figure 6. Association test between SNPs and APOBEC mutations.

There is no association between the 3 SNPs reported by Middlebrooks *et al.* (Nat. Gen. 2016) and the APOBEC mutations. The p values derived from Kruskal–Wallis test were shown in each panel.

Response Figure 7. Association test between SNPs and expression levels of A3A and A3B. There is no association between the 3 SNPs reported by Middlebrooks *et al.* (Nat. Gen. 2016) and the expression levels of A3A or A3B. The p values derived from Kruskal–Wallis test are present in each panel.

Response Table 2. Association test of the the 3 SNPs reported by Middlebrooks *et al.* (Nat. Gen. 2016) and the clinical indicators/A3B deletion polymorphism.

Lesion	rs10034748 (p=0.5981)			rs1004748 (p=0.2966)			rs1014971 (p=0.5981)		
	G/G	G/A	A/A	C/C	C/T	T/T	G/G	G/A	A/A
buccal mucossa	7	9	2	7	9	2	7	9	2
gingiva	3	2	0	3	2	0	3	2	0
gum	1	1	0	1	1	0	1	1	0
hard palate	0	2	0	0	2	0	0	2	0
mouth floor	0	1	0	0	0	1	0	1	0
retromolar trigone	1	2	0	1	2	0	1	2	0
tongue	13	6	0	12	7	0	13	6	0

Stage	rs10034748 (p=0.9344)			rs1004748 (p=0.8058)			rs1014971 (p=0.9344)		
	G/G	G/A	A/A	C/C	C/T	T/T	G/G	G/A	A/A
I	1	1	0	1	1	0	1	1	0
II	5	3	0	5	3	0	5	3	0
III	1	3	0	1	2	1	1	3	0
IVA	15	14	2	14	15	2	15	14	2
IVB	3	2	0	3	2	0	3	2	0

Gender	rs10034748 (0.146)			rs1004748 (p=0.1828)			rs1014971 (p=0.1456)		
	G/G	G/A	A/A	C/C	C/T	T/T	G/G	G/A	A/A
Male	24	18	2	24	17	3	24	18	2
Female	1	5	0	0	6	0	1	5	0

A3B	rs10034748 (p=0.4938)			rs1004748 (p=0.2071)			rs1014971 (p=0.4938)		
	G/G	G/A	A/A	C/C	C/T	T/T	G/G	G/A	A/A
-/-	6	2	0	6	2	0	6	2	0
-/+	10	14	1	9	15	1	10	14	1
+/+	9	7	1	9	6	2	9	7	1

4) The authors should evaluate and control for the infiltration of immune cells and stroma, which have been shown to affect the expression of APOBEC-family genes and correspond with clinical outcome (Leonard et al. CCR. 2016 and Smid et al. Nat. Comm. 2016). This should be done using both RNAseq and IHC to accurately assign expression of APOBEC family members to tumor/immune/stromal cells. In particular, IHC experiments using tissues from this unique cohort will unambiguously determine which cells are responsible for the abnormally high A3A signal (ie. is this the actual tumor cells or infiltrating macrophage lineage cells). Several commercial academic antibodies are available for these purposes.

Response:

Thanks for reviewer's suggestion. We used ESTIMATE reported by Yoshihara K, *et al.* (Inferring tumour purity and stromal and immune cell admixture from expression data. *Nat Commun* 4, 2612, 2013) to estimate the tumor purity and the fraction of stromal and immune cells from our 39 tumor/normal paired RNAseq data. There are significant differences in tumor/normal purity and immune score (Response Fig. 8a and 8b), but not the stromal score of tumor and normal samples (Response Fig. 8c). These results suggest there are infiltration of immune cells in our tumor samples. However, there is no difference in the immune score or stromal score between tumor samples carrying different A3B deletion genotypes, $A3B^{+/-}$ and $A3B^{-/-}$ (Response Fig 8d and 8e). Also we didn't find any correlation between the expression level of $A3A$ (or $A3B$) and the immune score or tumor purity in our tumor samples (Response Fig. 9). We also examined the APOBEC3A expression in representing tumor tissues from three genotypes individually by immunohistochemistry analysis using anti-APOBEC3A antibody (Sigma HPA043237). As presented in Response Fig. 10-12, we found the expression of APOBEC3A in both the OSCC tumor cells and the infiltrated immune cells, but not the stroma in tissue samples from all three genotypes. Taken together, even though there are infiltration of immune cells in our tumor samples, our analyses indicated that the $A3B$ deletion genotype but not the infiltrating immune cells leads to the differential expression of $A3A$ and different clinical outcome in Taiwan-OSCC.

Response Figure 8. ESTIMATE analyses of immune and stromal cells in paired samples. Significant difference was found for the estimated (a) presence of immune/stromal cells, and (b) immune score but not (c) stromal score between normal and tumor samples (Wilcoxon rank sum test). No significant difference was found for the estimated (d) immune score, and (e) stromal score from different genotypes in normal and tumor samples (Kruskal–Wallis test).

Response Figure 9. No significant correlation was found between the expression level of (a) *A3A* or (b) *A3B* and estimated tumor purity, stromal score and immune score. The p values derived from Spearman's rank correlation test are shown in each panel. The estimated tumor purity, stromal score and immune score are calculated from RNASeq with ESTIMATE.

Response Figure 10. Immunohistochemistry analysis of APOBEC3A in $A3B^{-/-}$ OSCC. Expression of A3A protein in $A3B^{-/-}$ OSCC tissue was detected by anti-APOBEC3A antibody. A3A was highly expressed in both tumor cells (T) and infiltrated immune cells (IC), but not in the stromal cells (SC) in a representing tissue. T, tumor cells; N, normal cells; IC, immune cells; SC, stromal cells.

Response Figure 11. Immunohistochemistry analysis of APOBEC3A in $A3B^{+/-}$ OSCC. Expression of A3A protein in $A3B^{+/-}$ OSCC tissue was detected by anti-APOBEC3A antibody. A3A was highly expressed in both tumor cells (T) and infiltrated immune cells (IC), but not in the stromal cells (SC) in a representing tissue sample. T, tumor cells; N, normal cells; IC, immune cells; SC, stromal cells.

Response Figure 12. Immunohistochemistry analysis of APOBEC3A in $A3B^{+/+}$ OSCC. Expression of A3A protein in $A3B^{+/+}$ OSCC tissue was detected by anti-APOBEC3A antibody. A3A was highly expressed in both tumor cells (T) and infiltrated immune cells (IC), but not in the stromal cells (SC) in a representing tissue sample. N, normal cell. T, tumor cells; N, normal cells; IC, immune cells; SC, stromal cells.

Minor comments

5) HO/HE/non nomenclature is confusing. $A3B^{-/-}$, $A3B^{+/-}$, $A3B^{+/+}$ would be much clearer.

Response:

Thanks for the reviewer's suggestion. We modified our nomenclature in the entire article: All HO, HE and non-carrier nomenclature was changed to $A3B^{-/-}$, $A3B^{+/-}$ and $A3B^{+/+}$ respectively in the revised text.

6) Subpanel designation (a, b, c, etc.) should be larger to more easily interpret figures.

Response:

Thanks for the suggestion. We have enlarged our subpanel designation.

7) Line 59. Should include a reference for how betel nut chewing is a risk factor.

Response:

Thanks for the suggestion. We added two references,

"In addition to the known risk behaviors of cigarette smoking and alcohol drinking, Taiwanese men often indulge in the additional risk behavior of betel nut chewing^{4,5}."

The newly added references are also included in the References section.

4. Ko, Y.C. *et al.* Betel quid chewing, cigarette smoking and alcohol consumption related to oral cancer in Taiwan. *J Oral Pathol Med* **24**, 450-3 (1995).
5. Sharan, R.N., Mehrotra, R., Choudhury, Y. & Asotra, K. Association of Betel Nut with Carcinogenesis: Revisit with a Clinical Perspective. *Plos One* **7**(2012).

8) Line 179. What about APOBEC3B peptides? Supplementary Fig 8. The peptide in the bottom row is identical in A3A, A3B, and A3G via protein BLAST. The peptide in the middle row is only 1 AA different from that seen most A3G entries in the nr protein database and is identical to the AA sequence used in an A3G structure (PMID: 25542899). Also an example of the raw data for the control should be shown.

Response:

We thank the reviewer to point out this issue.

(1) For A3B, we didn't detect any A3B-unique peptide in our proteomic experiments.

This may reflect the low abundance of A3B protein in samples. Regarding the last peptide, which is indeed shared by A3A, A3B and A3G. Therefore, we have removed that peptide from our revised Supplementary Fig 8 (shown below).

Supplementary Figure 8. A3A protein identification and quantification by LC-MS/MS analysis of OSCC tumors. Each panel shows both low mass reporter ion of iTRAQ reagent (114, 115 and 116) and MS/MS spectrum assigned to unique peptides of A3A in patient #1 (left 2 panels) and patient #22 (right 2 panels).

- (2) As for the peptide in the middle row of the original figure, one amino acid difference can indeed be differentiated with the mass spectrometry by peptide mass (m/z : 3468.4909 of A3A; m/z : 3512.4807 of A3G). Any change of amino acid residue in a specific peptide will result in changes in the peptide mass spectrum, which can be clearly differentiated by mass spectrometry. For example, BRAF with a single amino acid change was confidently identified by mass spectrometry in our recent report (PMID: 27497007).
- (3) In the paper published in PMID: 25542899, the authors introduce an amino acid mutation (D370A) in A3G for structure analysis, and the mutated peptide shared the same peptide sequence as A3A.
- (4) Each spectrum was contributed by all three labeled samples in iTRAQ technique. The iTRAQ technique quantifies peptides by labeling each sample with different isobaric tags (m/z : 114, 115, 116) and corresponding balance groups. For peptide identification by mass spectrometry in this study, the raw spectrum for the tissue-mixture control (labelled as 114 as controls), non-tumor tissue (labelled as 115), and OSCC tumors (labelled as 116) were all shown in the figure for each peptide. For example, at the Left half of the Top row, the left panel is the control for the peptide DAGAQVSIMTYDEFK identified in sample P01.

(5) As for the internal control proteins, to the best of our knowledge, there is no generally acknowledged protein expressed equally in non-tumor and tumor tissue of oral squamous carcinoma. In quantitative proteomics, we usually assume that most proteins remained unchanged, and perform a global normalization to normalize tumor/non-tumor (T/N) protein ratios across patients. It is worth mentioning, the normalized T/N ratios of commonly used internal controls (GAPDH, actin, and tubulin) in our proteomic data were unchanged (1.179 ± 0.369 , 0.817 ± 0.239 , and 1.224 ± 0.412 , respectively) (Response Fig. 10).

Response Figure 10. X Mass spectrum of GAPDH, actin and tubulin from in patient #1 and patient #22.

9) Line 189. APOBEC3A.

Response:

Thank you. We have modified the sentence at line 189

to “Given that the *A3A* and *A3B* were interferon-inducible²³⁻²⁷, this up-regulation may reflect the activation of interferon (IFN) signaling.”

The *A3A*, *A3B*, *A3C*, *A3D*, *A3F* and *A3G* genes were found to be interferon-inducible in the five reference we cited.

(1) *A3B*, *A3C*, *A3F* and *A3G* can be induced by interferon in ref. 23 “Bonvin, M. et al. Interferon-inducible expression of APOBEC3 editing enzymes in human

hepatocytes and inhibition of hepatitis B virus replication. *Hepatology* 43, 1364-1374 (2006)“.

- (2) *A3A*, *A3B*, *A3C*, *A3F* and *A3G* can be induced by interferon in ref. 24 “Peng, G., Lei, K.J., Jin, W.W., Greenwell-Wild, T. & Wahl, S.M. Induction of APOBEC3 family proteins, a defensive maneuver underlying interferon-induced anti-HIV-1 activity. *Journal of Experimental Medicine* 203, 41-46 (2006)”.
- (3) *A3A*, *A3B*, *A3C*, *A3E* (*A3D*), *A3F* and *A3G* can be induced by interferon in ref. 25 “Tanaka, Y. et al. Anti-viral protein APOBEC3G is induced by interferon-alpha stimulation in human hepatocytes. *Biochemical and Biophysical Research Communications* 341, 314-319 (2006)”.
- (4) *A3A* and *A3G* can be induced by interferon in ref. 26 “Peng, G. et al. Myeloid differentiation and susceptibility to HIV-1 are linked to APOBEC3 expression. *Blood* 110, 393-400 (2007)”.
- (5) *A3A*, *A3B*, *A3C*, *A3E* (aka *A3D*), *A3F*, *A3G* and *A3H* can be induced by interferon in ref. 27 “Koning, F.A. et al. Defining APOBEC3 Expression Patterns in Human Tissues and Hemaopoietic Cell Subsets. *Journal of Virology* 83, 9474-9485 (2009).”

10) Line 194. Authors should include these data. Also change wording to indicate that miRs destabilize mRNA transcripts rather than their absence stabilizes mRNA transcripts.

Response:

Thanks for the reviewer’s suggestion. We added this data into Supplementary Figure 10, and also add a paragraph to describe the reporter assay in our on-line methods. We tested the effect of has-miRNA-409 on the *A3A* 3’UTR and *A3B* 3’UTR. We found that miRNA-409 downregulated the *A3B* 3’UTR-linked but not the *A3A* 3’UTR-linked reporter gene expression.

Luciferase reporter assay

“*A3A* 3’UTR and *A3B* 3’UTR were PCR amplified from genomic DNA of OC3 oral cancer cell line, using the same forward primer 5'-ACTAGTAGGATGGGCCTCAGTCTCTAAG-3'; reverse primer 5'-AAGCTTAGTGTTTGTGGAAACTCTTGCAATT C-3' is for *A3A* 3’UTR, and 5'-AAGCTTAGTGTTTGTGGAAACAATTATGGAAG-3' is for *A3B* 3’UTR. The amplified PCR products were cloned into the pMIR-Report-Vector (Ambion). Precursor of miR-409 was amplified from OC3 cell genomic DNA, and cloned into pcDNA6.2-GW/EmGFP-miR (Invitrogen). 293T cells (2×10^5) were subjected to transient calcium-phosphate-mediated transfection with the following: 10 ng of pMIR-Luciferase-*A3B* 3’UTR; 1 μ g of the vector control or expression vectors encoding miR-409 and 10 ng of pCMV-Renilla (Promega). Luciferase and renilla

activities were measured using the Dual-Luciferase® Reporter Assay System (Promega). The luciferase values were normalized to those of renilla, and the results are presented as the luciferase/renilla ratio.”

The newly added Supplementary Fig. 10 is as follows.

Supplementary Figure 10. Potential miRNA regulation of *A3B* 3'UTR. (a) Luciferase reporter assay demonstrated that miR-409 may target the *A3B* 3'UTR but not *A3A* 3'UTR. Comparing to vector control (pcDNA6.2), ectopic miR-409 led to 20% reduction of luciferase activity ($p < 0.001$). (b) Schematic depiction of the predicted target site of miR-409 in the 3'UTR of *A3A_B*.

We also modified the sentence at lines 194-196 in our article.

From “We found that down-regulation of miR-409 in OSCC samples contributed to stabilizing transcripts carrying the *A3B* 3'UTR (data not shown).”

To “We examined the regulatory role of miR-409, one of these down-regulated miRNAs, and found that miR-409 can reduce the activity of luciferase reporter gene carrying *A3B* 3'UTR (Supplementary Fig. 10).”

11) Line 203. Proper statistical test should be applied to see if YTCW:RTCW ratio is significantly different. Also the cited paper used YTCA:RTCA ratio as an enrichment metric for APOBEC3A-mediated mutagenesis. Lastly these data should be subdivided by HO/HE groups.

Response:

Thanks for the reviewer’s suggestion. We carried out additional statistical tests to examine whether the differences between YTCA and RTCA/YTCW and RTCW are significant. The percentages of YTCA, RTCA, YTCW and RTCW in total mutation for our 50 samples do not follow normal distribution (Shapiro-Wilk test of normality, p value = 3.965e-07 and 0.0007816). Thus, we tested whether the percentages are different with Wilcoxon rank sum test and found they are significantly different (p -value = 0.0009131 and 1.704e-07 for YTCA vs. RTCA and YTCW vs. RTCW,

respectively). However, due to the small sample size, the difference was less significant or not significant when analyzed individually in three genotypes (8 $A3B^{-/-}$ samples and 17 $A3B^{+/+}$ samples in our cohort). We also found that there is no difference in the percentages of TCW, TCA, YTCA, RTCA, YTCW and RTCW in total mutations between different genotypes (Kruskal-Wallis rank sum test).

We modified the sentence as shown at lines 201-206 in the revised version.

“Since the hypermutation signature of $A3A$ may be distinguished from that of $A3B$ (YTCA vs. RTCA)³², we also examined their relative incidence in our samples. We found that both YTCA:RTCA and YTCW:RTCW ratios were about 7:3, and the number of YTCA/YTCW were significantly higher than RTCA/RTCW (Wilcoxon rank sum test, p value = 0.0009 and 1.704e-07 respectively, Supplementary Table 11), pinpointing $A3A$ as the major mutator contributing to the APOBEC-associated signature in OSCC.”

12) Line 229. This line makes it sound as if $A3B$ transcript is being detected. It should be clarified that this detection is due to mismapping of $A3A$ reads. Considering this high rate of cross mapping I would also like to see additional validation of unique $A3B$ and $A3A$ expression in this cohort. These reads and transcript levels should be excluded from analyses and graphs since they are clearly artifactual.

Response:

Thank the reviewer for pointing out this mis-mapping problem. To avoid mismapping of reads which come from $A3A_B$ 3'UTR to be mapped to the $A3B$ 3'UTR, we have re-analyzed the expression level of $A3B$ with proper genotype information.

We modified the sentence starting at line 227 to “As shown in Fig. 4b and Supplementary Fig. 13, $A3A$ expression was significantly elevated in tumor tissues compared to adjacent normal tissues from all three genotypes”.

We also add a sentence in the section “Identification of DEGs in OSCC samples from OSCC-Taiwan” of the on-line methods. “In order to accurately estimate the expression levels of $A3A$, $A3B$ and $A3A_B$, the genotype of each sample was taken into consideration for quantification.”

We modified the Fig. 4b with new results, which remain consistent with our original findings: $A3A$ was significantly upregulated in $A3B^{-/-}$, $A3B^{+/-}$ and $A3B^{+/+}$ samples, and $A3B$ was significantly upregulated in $A3B^{+/-}$ and $A3B^{+/+}$ samples.

We also added expression levels of *A3A* and *A3A_B* in Supplementary Fig. 13 (as shown below).

Supplementary Figure 13. Expression levels of *A3A* and *A3A_B* among three genotypes in OSCC tissue samples. Based on RNA-Seq-determined TPM values, the mRNA expression levels of (a) *A3A* and (b) *A3A_B* were determined in the initial cohort of 39 paired samples. Patients are grouped according to their APOBEC-deletion genotypes. ($A3B^{-/-}$: n=7; $A3B^{+/-}$: n=20; $A3B^{+/+}$: n=12)

14) Line 332. Logically this miRNA should also suppress the *A3A* transcript due to homology in the UTR.

Response:

Thanks for the reviewer's comment. We checked the 3' UTR sequence of *A3A* and *A3B*. TargetScan predicted that has-miR-409 targets at 3' UTR from *A3B* but not *A3A*. The seed region is at position 253-259 of *A3B* 3' UTR, and the sequence is "AACAUU". As shown in Response Fig. 11, the seed region (highlight in yellow) of 3' UTRs of *A3B* (AACAUU) has one-base mismatch with that of *A3A* (AAAAUU). The effect of has-miR-409 on 3'UTR was assayed by co-transfection of the expression vector of has-miR-409 and the reporter gene carrying *A3A* 3'UTR or *A3B* 3'UTR in cells. Results showed that has-miR-409 suppressed the reporter gene

carrying *A3B* 3'UTR but not the one linked to *A3A* 3'UTR as measured by the luciferase activity.

CLUSTAL multiple sequence alignment by MUSCLE (3.8)

```

A3A 3' UTR      AGGAUGGGCCUCAGUCUCUAAGGAAGGCAGAGACCUGGGUUGAGCAGCAGAAUAAAAGAU
A3B 3' UTR      AGGAUGGGCCUCAGUCUCUAAGGAAGGCAGAGACCUGGGUUGAGCAGCAGAAUAAAAGAU
*****
A3A 3' UTR      CUUCUCCAAGAAAUGCAAACAGACCGUUCACCACCAUCUCCAGCUGCUCACAGACGCCA
A3B 3' UTR      CUUCUCCAAGAAAUGCAAACAGACCGUUCACCACCAUCUCCAGCUGCUCACAGACACCA
*****
A3A 3' UTR      GCAAAGCAGUAUGCUCGCCGAUCAAGUAGAUUUUUAAAAAUCAGAGUGGGCCGGGCGCGG
A3B 3' UTR      GCAAAGCAAUGUGCUCUGAUCAAGUAGAUUUUUAAAAAUCAGAG-----
***** *
A3A 3' UTR      UGGCUCACGCCUGUAAUCCAGCACUUUGGAGGCCAAGGCGGGUGGAUCACGAGGUCAGG
A3B 3' UTR      -----
A3A 3' UTR      AGAUCGAGACCAUCCUGGCUAACACGGUGAAACCCUGUCUCUACUAAAAAUACAAAAAU
A3B 3' UTR      -----
A3A 3' UTR      UAGCCAGGCGUGGUGGGCGGCCUGUAGUCCAGCUACUCUGGAGGCUGAGGCAGGAGA
A3B 3' UTR      -----
A3A 3' UTR      GUAGCGUGAACCCGGGAGGCAGAGCUUGC GGUGAGCCGAGAUUGCGCUACUGCACUCCAG
A3B 3' UTR      -----
A3A 3' UTR      CCUGGGCGACAGUACCAGACUCCAUCUCAAAAAAAAAAAAAACCAGACUGAAUUUUUUUA
A3B 3' UTR      -----UCAAUUUUUUUA
* *****
A3A 3' UTR      ACUGAAAAUUUCUCUUAUGUCCAAGUACACAUAAGUAAGAUUAUGCUCAAUAUUCUCAG
A3B 3' UTR      AUUGAAAAUUUCUCUUAUGUCCAAGUGUACAAGAGUAAGAUUAUGCUCAAUAUUCUCCAG
* *****
A3A 3' UTR      AAUAAUUUUCAAUGUAUUAAUGAAAUGAAAUGAUAAUUUGGCUUCAUAUCUAGACUAACA
A3B 3' UTR      AAUAGUUUUCAAUGUAUU-----AAUGAAGUGAUUUAAUUGGCUCCAUAUUUAGACUAUA
**** *****
A3A 3' UTR      CAAAUUUAAGAAUCUCCAUAUUUGCUUUUGCUCAGUAACUGUGUCAUGAAUUGCAAGAG
A3B 3' UTR      AAACAUUUAAGAAUCUCCAUAUUUGCUUUUGCUCAGUAACUGUGUCAUGAAUUGCAAGAG
** ***** *
A3A 3' UTR      UUCCACAAACACU
A3B 3' UTR      ----CACUAGCAAA
*** * **

```

Response Figure 11. Sequence alignment result for the 3'UTRs from *A3A* and *A3B*. The seed region of predicted has-miR-409 binding site “AACAUU” in *A3B* was highlighted in yellow together with the homolog sequence in *A3A* 3'UTR. The one mismatch in seed region is highlighted in red.

15) Lines 331 & 334 Why are the data not shown?

Response:

Thanks for reviewer's suggestion. We added the miRNA data as the Supplementary Fig. 10, and modified the sentence at Line 332

“Interestingly, most of these miRNAs were down-regulated in oral cavity cancer in TCGA dataset.”

And line 334,

To “This assay showed that the over-expression of miR-409 reduced *A3B* 3'UTR reporter activity by 20% (Supplementary Fig. 10)”

16) Fig 3d should be removed from main figure set. It does not add valuable information to the paper, nor is adequately addressed for a mechanistic explanation for A3A contributing to cancer mutagenesis in this manuscript.

Response:

We moved fig. 3d to Supplementary Figure 9, and modified the sentence at line 190 to “Indeed, our transcriptome-sequencing data are consistent with this hypothesis (Supplementary Fig. 9 and Table 9).”

17) Fig 4a-d. This is a lot of validation and can be more concisely summarized and mostly moved to a supplemental figure.

Response:

Thanks for reviewer's suggestion. We combined the original Fig. 4a and 4c as Fig. 4a and moved Fig. 4b and 4d to the Supplementary Figure 12. We modified the sentence at line 216 to “In our 50 OSCC matched samples, we detected germ-line deletion of the *A3B* coding sequence in 33 individuals: eight *A3B*^{-/-} individuals (homozygous for the deletion allele) and 25 *A3B*^{+/-} individuals (heterozygous for the deletion allele) (Supplementary Fig. 12a). The exome sequencing results were confirmed by PCR using genotype-specific primer sets that distinguished among the *A3B*^{-/-} (a 757-bp PCR product), *A3B*^{+/-} (449-bp and 757-bp), and *A3B*^{+/+} (449-bp) genotypes (Fig. 4a right). Our RNA-Seq data further substantiated the presence of a variant transcript corresponding to the genomic polymorphism (Supplementary Table 12). RT-PCR analysis with fusion-sensitive primers independently confirmed the expression of the deletion variant in *A3B*^{-/-} and *A3B*^{+/-} samples (Supplementary Fig. 12b).”

Line 236,

To “Among these individuals, *A3A* expression was significantly elevated in all tumor tissues, with higher levels seen in *A3B*^{-/-} individuals versus those with the *A3B*^{+/-} or *A3B*^{+/+} genotypes (Fig. 4c, left). In contrast, *A3B* expression levels were higher in the tumor tissues of *A3B*^{+/-} and *A3B*^{+/+} genotypes compared to normal tissues, whereas it was not detected in *A3B*^{-/-} genotype tumors as expected (Fig. 4c, right).”

And line 276 to “Of the 188 patients with clinical outcome data, we were able to genotype 143 (Fig. 4c).”

The revised Fig. 4 is as follows.

Figure 4. The deletion polymorphism of A3A-A3B genomic locus and upregulation of A3A and A3B in OSCC-Taiwan. (a) Schematic depiction of the gene structures and genomic organizations of the deletion polymorphism (top) and non-deletion (bottom) versions of the *APOBEC3A-APOBEC3B* genomic locus. The deletion variant (*APOBEC3A_B* genotype) arose from a 29.5-kb genomic deletion spanning from the 3'UTR of *A3A* to the eighth exon of *A3B*. PCR-based genotyping analysis was used to distinguish non-deletion and deletion alleles according to the size of the amplified product (449 bp and 757 bp, respectively). *A3B*^{+/+}, *A3B*^{+/-} and *A3B*^{-/-} represent for non-carrier, heterozygous and homozygous for deletion alleles, respectively. (b) Genotype-biased expressional alterations of *A3A* and *A3B* in OSCC. Based on RNA-Seq-determined TPM values, the mRNA expression levels of *A3A* (left) and *A3B* (right) were determined in the initial cohort of 39 paired samples. Patients are grouped according to their APOBEC-deletion genotypes, and n, the number of patients in each group, are indicated on the top. (c) Genotype-specific relative expression of *A3A* in tumor versus normal tissue samples, as determined by RT-PCR. Individuals of the *A3B*^{-/-} genotype exhibited a greater tumor-specific up-regulation of *A3A* than those of the *A3B*^{+/-} genotype ($p = 0.0354$) (*, $p < 0.05$; **, $p < 0.001$; ***, $p < 0.0001$).

The new supplementary Fig. 12 is present as follows.

Supplementary Figure 12. Detection of fused gene *A3A_B* in WES, RNASeq and RT-PCR. (a) Representative distribution of RNA-Seq and WES reads in the APOBEC3 locus from *A3B*^{-/-} (top), *A3B*^{+/-} (middle), and *A3B*^{+/+} (bottom) samples. Reads for RNA-Seq and WES are shown in green and blue, respectively. Red lines represent reads corresponding to junctional sequences. The height of a peak is proportional to the read number. (b) RT-PCR of the *A3A_B* fusion transcript. The 800-bp RT-PCR product was derived from the *A3A_B* fusion transcript. PCR products were resolved by Sanger sequencing (top) or size (gel electrophoresis, bottom) to distinguish genotype-specific transcript expression.

18) Why in Fig 5 do the *A3A* low expressers in HE and HO have drastic survival differences. Wouldn't it be expected if this was due to *A3A* that low expression would be nearly identical?

Response:

Based on the results of our study, the higher expression level of *A3A* is associated with better overall survival in patients carrying *A3B*^{+/-} and *A3B*^{-/-}. In this study, we chose the median of *A3A* values in all tested samples as the cutoff value. Thus, in those with poorer overall survival should all be the ones with *A3A* with lower than the

cutoff value in either $A3B^{+/-}$ or $A3B^{-/-}$. The previous reports indicated that the $A3B$ deletion may be a risk factor for breast and ovarian cancers (Xuan et al., Carcinogenesis 2013, Qi et al., Tumour Biol 2014 and Wen et al., Breast Cancer Res 2016). Thus, whether the $A3B$ deletion polymorphism alone can be a potential predictive biomarker for OSCC in Taiwan ought to test in a larger cohort.

19) Supplementary methods p21. Both primers for A3B are referred to as A3B-qrtF.

Response:

Thank you for pointing-out the mistake. The second primer should be A3B-qrtR. We made correction in the revised on-line methods.

20) Supplemental fig S9 should also show correlation coefficient as well as highlight all samples in this study by A3B deletion status and any other impactful mutations (such as those in p53).

Response:

Thanks for the reviewer's comments. We have highlighted the A3B deletion status in the plot: green dots represent $A3B^{+/+}$, blue dots represent $A3B^{+/-}$, and red dots represent $A3B^{-/-}$. Please see the Supplementary Fig. 11 (also shown below) in the revised version, which was the Supplementary Fig S9 in the original manuscript. As noted, samples from all 27 patients in the plot harbored TP53 mutations.

Supplementary Figure 11. Correlation between APOBEC-associated single nucleotide variations (SNVs) and expression level of *A3A* and *A3B*. The Y-axis represents the number of somatic TCW (C>T, C>G) mutations, while the X-axis shows the expression level of (a) *A3A* and (b) *A3B* on the basis of TPM (transcripts per million). The p values were calculated with Spearman's rank correlation. The dots colored by green represent $A3B^{-/-}$, the dots colored by blue represent $A3B^{+/-}$, and the dots colored by red represent $A3B^{+/+}$.

21) Samples with <200X coverage and ~40% reads mapped to target are questionable for such a consistently high coverage HiSeq run. Do these look like outliers in the data analysis?

Response:

Thanks for reviewer's kind reminder. There are 16 samples having sequencing depth less than 200X (145X~197X) in either tumor or normal tissues. To test if these samples were outliers, we checked (1) the proportion of mutation types including SNVs and InDels, and (2) the total number of mutations of these <200X coverage or ~40% on-target-rate samples. We found there was no difference between these samples and other samples. The results are as follows.

We compared the six different mutation types and total number of mutations across all our 50 samples (Response Fig. 12a and Fig. 13a). We further compared the

proportion of mutation types between low coverage and high coverage samples, and we found no difference (Response Fig. 12b and Fig. 13b). We carried out similar analysis for the 8 samples having relatively lower on-target rates (44.96% ~ 49.39%) in either tumor (Response Fig. 12) or normal part (Response Fig. 13). We found no difference in mutation profile between samples having low or high coverage. Thus, we concluded that these low coverage or low on-target-rate samples are not outliers.

Response Figure 12. Mutation profiles in all 50 samples based on on-target rate in tumor tissues. (a) Mutations in all 50 samples. In the top panel, all 50 samples were sorted based on the coverage depth of tumor samples by NGS and the red color bar represents samples having coverage depth less than 200X. The middle panel shows the proportion of indels and types of base substitution mutations. The bottom panel illustrates the total number of mutations. The six types of SNVs and InDels were coded with different colors. (b) The average proportions mutations for low and high depth samples; error bars denote 1X standard deviation.

Response Figure 13. Mutation profiles in all 50 samples based on the on-target rate of PBMC samples. (a) Mutations in all 50 samples. In the top panel, 50 samples were sorted by the depth of normal PBMC samples and the red color bar represents samples having coverage depth less than 200X. The middle panel shows the proportion of indels and types of base substitution mutations. The bottom panel illustrates the total number of mutations. The six types of SNVs and InDels were coded with different colors. (b) The average proportions mutations for low and high depth samples; error bars denote 1X standard deviation.

22) The authors should also consider subsetting TCGA samples by cancer stage to better compare to their cohort, which is >60% late-stage cancer (4A).
Response:

Thanks for the reviewer’s suggestion. Among 314 TCGA oral cavity cancer samples, 217 provided the pathological stages including 55 stage III, 155 stage IVa and 7 Stage IVb, respectively. We used these late-stage samples for the survival analysis and

found no significant association between *A3A* or *A3B* expression levels and overall survival. We also tried used only Stage IVa samples (n=155) still we found no significant association between *A3A* or *A3B* expression levels and overall survival in these samples. The Kaplan-Meier plot (Response Fig. 14).

Response Figure 14. Kaplan-Meier plot for survival analysis by *A3A* and *A3B* expression for late-stage samples in TCGA. There is no significant association between *A3A* (left) or *A3B* (right) expression levels and overall survival in (a) late-stage (stage III + stage IV) or (b) stage IVa OSCC-TCGA samples. The samples were divided into high expression level of *A3A* (or *A3B*) based on the median expression value.

Reviewer #2 (Remarks to the Author):

Chang et al describe a germline variant of the APOBEC genes A3A and A3B in the Taiwanese population, and show that this seems to have consequences for the mutational profile in oral cancers. The authors did a lot of work. The findings are interesting but some critical information is still missing.

Comments:

1) Is Figure 1 not mislabeled with respect to deletion and amplification (copy number)? Usually CDKN2A is deleted and EGFR amplified. Please correct and check labels everywhere. Should the Apobec cases not be identified in the same Fig? There are 4 HPV cases identified, which all have a TP53 mutation. This is highly unlikely. How was HPV status defined, only by NESTED E6 RT-PCR. Very unreliable. E6 RT-PCR on RNA should be used. Alternatively, the authors have RNAseq data and exome data and should map the reads against HPV. My guess is that they are all negative. If they are truly positive these cases should be marked in the subsequent analyses as these are typically Apobec-type tumors.

Response:

Thanks for pointing out the mislabeling in the Fig. 1. We have made correction in the revised Fig. 1c. We also added the genotype information of each sample (patient) at the bottom of the revised Fig. 1d. For generation of our RNA sequencing and whole exome sequencing data, the libraries were prepared with capture kits. As the capture kits are designed for analysis of human exome, we could not extract HPV-related information from sequencing experiments. In this report, the HPV DNA was amplified by PCR using genomic DNA prepared for exome sequencing, and RT-PCR using cDNA samples prepared for RNAseq experiments. The results of PCR and RT-PCR are summarized in Response Table 3.

Response Table 3. HPV status checked with results of PCR and RT-PCR in samples.

Sample #	HPV-DNA *	HPV-E6 cDNA
25	type 18	undetectable
56	type 16	undetectable
51	type 16	undetectable
49	type 16	undetectable

*HPV-DNA PCR products were confirmed by Sanger's sequencing.

The authors used the Lawrence filter on their data in Sup Fig 6, but present unfiltered mutations already in Sup Fig2. I noted TTN as frequently mutated gene which is a typical suspect. They should warn in Sup Fig 2 that this is unfiltered data. Which VAF was used to call a mutation?

Response:

Thanks for the reviewer's suggestion. We have added one sentence in the figure legend of Supplementary Fig 2a: "This figure shows affected genes without applying variant allelic fraction filters". We used Mutect and Indelocator to call SNV and InDels, respectively, with their default parameter setting. Both Mutect and Indelocator have their own statistical model to determine the minimum variant allele fractions.

2) The authors define that 38% of the cases showed an Apobec signature, but this is critical information and should be displayed in Figure 2 per case, including how it was defined. Just an algorithm score? How did these scores vary, also in respect with HO or HE? See also point 4.

Response:

Thanks for the reviewer's suggestion. The percentage of each signature was calculated by the contributions of mutations of signatures deciphered by Wellcome Trust Sanger Institute (WTSI) Mutational Signature Framework with default parameters (Cell Reports 3, 246-259, January 31, 2013). The percentage of each signature is calculated by the total 20,963 somatic mutations of all samples, which are assigned in to each signature by WTSI framework. We have modified the sentence at line 158 to "By applying a similar algorithm to data for the 20,963 somatic SNVs discovered herein, we identified three mutational signatures as being enriched"

Response Fig. 15 showed the contribution of mutation signatures in all 50 samples. After re-calculated the percentage of mutation signatures, the APOBEC signature is about 40%, whereas those of age and smoking are 37% and 23%, respectively (Response Fig. 15). The percentage of each signature was slightly different from those reported in the original manuscript. This is due to that we did not calculate mutations in the negative strand in our original version. We revised all the results accordingly.

Response Figure 15. The contributions of signatures of each genome in 50 OSCC- Taiwan. The X-axis shows individual samples, and Y-axis shows the number of mutations contributed by APOBEC (red), age (orange) and smoking (blue) according to the WTSI framework.

3) Sentence 168 and 169 are very confusing. The authors state that Apobec is typically OSCC while it is absent in the Indian OSCC cohort?

Response:

Thanks for the reviewer’s comments. In fact, APOBEC signature is also found in OSCC-India but to a lesser extent (17%). We rephrased our sentences at line 165 to “Notably, the APOBEC-associated signature was highly represented in the OSCC-Taiwan and OSCC-TCGA datasets, but to a lesser extent (17%) in the OSCC-India and HNSC-TCGA (31%) datasets (Fig. 2c and Supplementary Fig. 6)”

4) I miss a Figure that associates the Apobec score of the tumor against the genotype HO, HE or non. There are only associations with A3A and A3B expression while the key point is the association of genotype (and HPV status) with Apobec score.

Response:

Thanks for the reviewer’s suggestion. We examined whether there are differences in TCW mutation rate in different A3B deletion genotypes or HPV status. From our 50

whole exome sequenced samples, we did not find significant difference of the percentages of TCW in total mutation between samples having different A3B deletion genotypes or HPV infection status (Response Fig. 16). Given that we found the differentially higher expression of A3A in A3B^{-/-} and A3B^{+/-} than in A3B^{+/+} genotypes, and a significant correlation between TCW mutation and A3A expression level, it should be worth for further examination with larger datasets.

Response Figure 16. Association test between samples having different (a) A3B deletion genotypes and (b) HPV infection status. The *p* values derived from Kruskal–Wallis test and Wilcoxon rank sum test were shown with box plots.

5) The Apobec genotype is in the germline and not somatic. Hence a difference in expression between tumor and normal is not to be expected. The authors suggest that this is related with the A3B 3'UTR and aberrant microRNA expression in the tumor. An interesting hypothesis but it should be proven in a functional experiment and shown that the fusion gene is indeed expressed at a higher level in tumor cells than in normal cells. Obviously this requires some reference transfection by a marker gene, but it is do-able as mucosal keratinocytes can be cultured and transfected.

Response:

Thanks for the reviewer's suggestion. The differential expression of A3A between tumor and its normal part strongly suggested the A3A expression is induced in the tumor tissues. We hypothesized that at least two levels of gene regulation, both transcriptional and post-transcriptional are involved. (1) APOBEC genes are inducible by IFN signaling pathway, which is highly elevated as presented in our RNAseq data

from the tumor tissues (also in Supplementary Figure 9). In contrast to this tumor-associated up-regulation, *A3A* expression levels in normal tissues of all three genotypes were significantly low. (2) hybrid *A3A_B* is found more stable than *A3A* by Cavel, *et al.* (A prevalent cancer susceptibility APOBEC3A hybrid allele bearing APOBEC3B 3'UTR enhances chromosomal DNA damage. *Nat Commun* **5**, 5129 2014), suggesting that miRNAs may be involved. We examined this possibility by predicting the miRNAs that can target *A3B* 3'UTR and found most of those miRNAs are down regulated in oral cavity cancer in TCGA database. We validated one of them, has-miR-409, with *A3B* 3' UTR reporter assay (Supplementary Fig. 10). Please also see the paragraph “**Taiwan-OSCC-associated up-regulation of *A3A* expression**” in the Discussion section of the revised version.

6) The clinical association study should be described in much more detail. How were patients treated? Were all margins histologically tumor-negative? How were recurrences and particularly second primary tumors evaluated? How frequent was the followup?

Response: We appreciate the reviewer’s comment, and the detailed description of the clinical information has been added to the on-line methods of the revised manuscript. “The OSCC primary tumors were all excised with adequate surgical margins, and the tumor margin tissue was sent for intraoperative fresh frozen section histopathology analysis. If margins were not deemed to be tumor free, further resection was performed. Nonetheless, five of the 188 resected OSCC specimens were found to be tumor-positive in surgical margins. Various types of neck dissection were performed according to the primary tumor site and clinical lymph node status. Postoperative radiotherapy was performed on patients with pathologically identified T4 tumors and positive lymph nodes within 6 weeks following surgery. Patients with any of the following pathological features received adjuvant concurrent chemoradiotherapy: metastasis in multiple neck lymph nodes, extracapsular spread, positive surgical margins, perineural invasion or nodal dissemination at level 4 or 5. The chemotherapy was a cisplatin-based regimen, and the total radiation dose was 66 Gy. The prescribed dose was delivered in fractions of 1.8–2-Gy per day for 5 days per week. Patients underwent standard postoperative work-ups according to institutional guidelines, which included complete physical examination at regular follow-up visits. Computed tomography or magnetic resonance imaging of head and neck and chest radiographs were performed 3 months after the treatment and every 6 months for 3 years. Additional radiological examinations, bone scans, and abdominal ultrasonography were performed for any suspicious recurrence or second primary tumors noted clinically. All patients completed regular follow-up visits every 2–3

months for the first year after discharge, every 3–4 months for the second and third years, and every 6 months thereafter.”

Reviewer #3 (Remarks to the Author):

Chen et al explore the specific relationship between a common deletion allele which results in the fusion of APOBECA and B genes (deletion of A3B) in the Taiwanese population with oral cancer. This is a particularly interesting cancer to study given the unique environmental factors that influence OSCC (betel nut chewing) as well as the increasing knowledge of mutation patterns and is clinically very significant given the high morbidity and mortality. The authors highlight that a frequent germline polymorphism which results in fusion of APOBEC3A and the 3'UTR of APOBEC3B is frequent in some populations (including the Taiwanese) and may influence the biology of the tumor development. The authors also provide some initial data that presence of this polymorphism (particularly when homozygous) may also influence survival. This study highlights the increasing complexity of understanding cancer genomics as these studies extend from the initial cohorts of European whites to many different global populations.

Major

1. The abstract is very hard to follow if someone isn't familiar with the gene cluster. It would be good to clarify the source of the fusion transcripts from the deletion as well as the patient population being studied (and how many have deletion alleles). The abstract also doesn't convey that there are two different independent sets being studied.

Response:

Thanks for reviewer's suggestion. We have modified the abstract.

Oral squamous cell carcinoma (OSCC) is a prominent cancer worldwide, particularly in Taiwan. By integrating omics analyses in 50 matched samples, we uncovered APOBEC-associated mutation signatures as a predominant variant, correlating with the elevated expression of *APOBEC3A* in the *APOBEC3* gene cluster at 22q13. *APOBEC3A* expression was significantly higher in tumor tissues carrying *APOBEC3B*-deletion allele(s), where *APOBEC3A_B* fusion transcript was identified. High-level *APOBEC3A* expression was associated with better overall survival especially among patients carrying *APOBEC3B*-deletion alleles as examined in a second cohort (n=188; $p=0.004$). The frequency of *APOBEC3B* deletion alleles is about 50% in 143 genotyped OSCC-Taiwan samples ($27A3B^{-/-}:89A3B^{+/-}:27A3B^{+/+}$), compared to the 5.8% found in 314 OSCC-TCGA samples. We thus report the first example of a germ-line-associated prognosis biomarker in OSCC, and introduce a

hitherto unknown genetic component of this disease. Our finding is a paradigm that might be repeated with other biomarkers and polymorphisms in other cancer types.

2. For any paper describing both germline and somatic mutations it is very helpful to be very clear when you are referring to each type of data. For example, is the following sentence referring to unique somatic mutation pattern or germline variation from the local population. It appears from the figure legend to be somatic but it would be helpful to clarify in the text.

“Interestingly, however, our analysis further identified mutations in seven genes (LCE4A, ORAI1, ZNF717, PABPC3, SKA3, ODF1, and SETD8) that turned out to be unique, locally prevalent ($\geq 10\%$ of patients) “

Response:

Thanks for the reviewer’s suggestion. We have revised the sentence in the revised text by adding “somatic” mutation instead of mutation. We also removed the InDels identified by VarScan2, and the modified the sentences (lines 129-134) in the revised manuscript to “We found that our patient cohort was characteristic of OSCC with regard to frequently mutated genes, such as *TP53*, *FAT1*, *NOTCH1*, *PIK3CA*, *CDKN2A* and *HRAS*. Interestingly, however, our analysis also identified somatic mutations in genes that turned out to be unique (*CENPV*) or locally prevalent (*DHRS4*, *RASAI*, and *SETD8* in $\geq 10\%$ of patients) (Fig. 1b, and Supplementary Table 6 and Supplementary Fig. 3).”

3. It is an important point that the gene expression of A3B detected in some studies results from transcripts from the 3’UTR. A comment in the Discussion to researchers doing expression analysis to not assume that a gene is over-expressed without being clear that the coding region itself is being over-expressed would be useful.

Response:

Thanks for the reviewer’s suggestion. We had added a new paragraph in the Discussion section.

“To our knowledge, many quantification methods tend to utilize expectation-maximization algorithms to estimate the maximum likelihood expression levels for multiple mapped reads. However, one should pay special attention to the application of RNAseq data for correct quantification of the expression levels for *A3A*, *A3A_B* and *A3B* in samples carrying the *A3B* deletion genotype. Given that *A3A_B* has exactly the same 3’UTR sequence as *A3B*, the reads to *A3A_B* could be mis-mapped to those for *A3B*, resulting in the reads assigned to *A3B* in *A3B*^{-/-} samples. Notably, *A3A_B* was not annotated in human genome references and thus may not be considered in transcriptome quantification analysis in earlier reports. Thus, our findings further emphasize the

importance of cross-examination of expression data with the genomic information.”

4. In the second set of tumors, the authors state that all of the tumors had over-expression of A3A or A3B. However, the mutation pattern associated with APOBEC was only seen in 38% of the first cohort. Given the frequency of the A3B deletion allele in the larger set why is it only a minority that have this mutation spectrum? Similarly, the authors never comment on whether these alleles are enriched in this tumor population versus the ethnic Taiwanese population as a whole. This should be clearly stated (or the reason it isn't studied provided).

Response:

[Redacted]

[Redacted]

5. For Figure 4 panel E – do the authors want to discuss why they think there is such a wide range of expression of both A3A and A3B in the tumor samples.? Since this is stated to be done by RT-PCR, why couldn't the authors use a PCR that would not detect the deletion allele?

Response:

Thanks for the reviewer's suggestion. In our study, Fig. 4b (the original Fig. 4e) shows the expression level of *A3A* and *A3B* in genotype of *A3B*^{-/-}, *A3B*^{+/-} and *A3B*^{+/+} by NGS data analysis. We validated NGS results using qRT-PCR method (Fig. 4c i.e. the original Fig. 4f). To make sure the *A3B* gene expression level in Fig. 4b is contributed by only *A3B* but not the 3'UTR of *A3A_B*, we re-analyzed our NGS data (for details, please refer to the response of question 12 from reviewer 1). The primers of qRT-PCR used to differentiate *A3A* and *A3B* are presented in the Response Fig. 17. Using *A3A* specific primers A3A_q2F and A3A_q2R to generate 176 bp of qRT-PCR products (from either *A3A* or *A3A_B*), which can be detected in all three genotypes of clinical samples; however, the 256-bp qRT-PCR product specific to *A3B* generated by primers A3B_qRTF and A3B_qRTR was only detected in samples of *A3B*^{+/+} and *A3B*^{+/-} genotypes. The unique *A3A* and *A3B* qPCR products are confirmed by Sanger Sequencing (please refer to the response to Reviewer 1, question 2). The results of NGS were agreeable with those from qRT-PCR in all samples tested, further confirmed the wild-range expression levels found in the tumor samples, which may reflect the heterogeneous property of clinical samples.

Chr:22q13.1

A3B deletion polymorphism

Response Figure. 17. Schematic depiction for the genomic organization of the deletion (top) and non-deletion (bottom) alleles of *A3A* and *A3B* genes at chr. 22q13.1. The primers were designed to differentiate 3 forms of transcripts; *A3A_B*, *A3A*, *A3B*. The primers designed for *A3A* can detect transcripts of both *A3A* from non-deletion allele and *A3A_B* from deletion allele. The primers designed for *A3B* is located outside 3'UTR and can only detect *A3B* expression but yielded no PCR products from the deletion allele.

6. Why do you think that being heterozygous for the deletion allele isn't enough to give increased expression of A3A? Given that there are three genotypes why did you decide to use just two categories (high and low) for the survival analysis. Did the expression cut-off used mirror what you see with the homozygous deletion allele carriers or include the heterozygous carriers?

Response:

Given that both *A3A* and *A3B* can deaminate the cytidines of DNA and *A3A* has much higher activity than *A3B*, there is likely a regulatory mechanism for the expression levels of *A3A* and *A3B*. Therefore, the expression level of *A3A* may actually under tight constrain as long as there is still one copy of *A3B* in *A3B*^{+/-} individuals. When we analyzed the correlation between the expression level of *A3A* (or *A3B*) and clinical outcome, we chose the median value that is considered as the most unbiased cutoff for the survival analysis. The same cut-off was used to stratify homozygous deletion allele carriers or the heterozygous carriers in Fig 5c. Since the sample size for each genotype was considerably small, it would be hard to make statistically relevant results. We agree with the reviewer's thoughts. This should be further tested in a large cohort with sufficient sample numbers from each genotype.

7. The survival data is being assessed from a relatively small cohort and it isn't

clear if multi-testing was accounted for in all the different analyses being done. The authors may want to make a statement that the survival data would need to be replicated in a larger cohort.

Response:

Thanks for reviewer's suggestion. We agree with the reviewer's concern. The current study revealed the association of A3A expression level and better overall survival in a relatively smaller cohort. These findings need to be replicated in a larger cohort. We have added sentences in the first paragraph of the Discussion section of the revised version to address this issue.

“Our findings indicate a strong association of the A3B deletion genotype with A3A expression and clinical outcome. Given the APOBEC-induced mutations are now known prevalent in many cancers, it is worth to validate this association in a larger cohort or in other cancer types in the future.”

Minor:

1. Which “Americans” are being referred to in the abstract. Do you mean Amerindians? The white US population typically resembles European in frequency.

Response:

Thanks for the reviewer's correction. It should be Amerindians.

2. For the international audience of Nature Communications it would be helpful to describe the origins of “ethnic Taiwanese”

Response:

Thanks for the reviewer's suggestion. We have modified a sentence in the Discussion section of the revised version at line 302 to “This is the first report of APOBEC deletion polymorphisms in a Taiwanese population, mainly Han-Chinese.”

3. You may want to comment in the Discussion that almost all of the patients also have a history of cigarette smoking in addition to betel nut chewing.

Response:

Thanks for reviewer's suggestion. We have modified sentences in our Discussion at line 369 to “Betel nut chewing is a regional risk factor for OSCC, and 40 out of the 50 OSCC patients recruited in the discovery cohort had a history of both betel nut chewing and cigarette smoking.”

line 377 to “The overwhelming majority (80-90%) of our study subjects in both cohorts reported chewing betel nut. Thus, ...”

REVIEWERS' COMMENTS:

Reviewer #1 (Remarks to the Author):

The authors have done a very nice job addressing the extensive comments of all 3 reviewers. In particular, Fig 1 now includes most of the relevant information along with connections to APOBEC and HPV genotypes. As long as the WES and RNAseq data sets are publicly accessible, I am supportive of publication.

The paper is generally well written but could still benefit from professional editing (ex. disc: "Given the APOBEC-induced mutations are now known prevalent in many cancers, it is worth to validate this association in a larger cohort or in other cancer types in the future." should be "... it may be worthwhile testing/checking (not validating) this association in larger cohorts and other cancer types ...").

Response:

Thanks for the reviewer's comments and suggestion. As described in our on-line methods, the WES and RNASeq data were all uploaded to NCBI Sequence Read Archive (SRA) under the accession number SRP078156 and will become public accessible after this manuscript published.

We have sent the paper for professional editing work as suggested by the reviewer. Please also check the last sentence in the first paragraph of the Discussion section (line 317-319).

Reviewer #2 (Remarks to the Author):

Chen et al revised their MS considerably, and it really improved. Nonetheless I still have some issues that have not been addressed.

- I had a remark before on the HPV status. Chen et al show in their reply that all cases were negative for E6 expression, Moreover 3 of 4 have p53 mutations, typically not related to HPV. All these cases are not driven by HPV in my view, which is good in the context of this subject as HPV tumors show APOBEC signatures. Hence it should be removed from Fig 1. The authors can indicate in the M&M or Suppl information that they tested for HPV DNA and RNA, which cases were positive for DNA, but that they did not consider them as HPV driven and why.

Response:

Thanks for the reviewer's suggestion. We have removed the information for HPV infection. Here is the revised version of Figure 1.

Chen et al show in their reply that all cases were negative for E6 expression, Moreover 3 of 4 have p53 mutations, typically not related to HPV. All these cases are not driven by HPV in my view, which is good in the context of this subject as HPV tumors show APOBEC signatures. Hence it should be removed from Fig 1. The authors can indicate in the M&M or Suppl information that they tested for HPV DNA and RNA, which cases were positive for DNA, but that they did not consider them as HPV driven and why.

Response:

Thanks for the reviewer’s suggestion. We added several sentences from line 207-210. “Four out of the 50 cases were found to be HPV DNA-positive. However, three of them carried *TP53* mutations and all four were E6 transcript-negative (Supplementary Table 12)³³⁻³⁵, suggesting that the mutation signatures were unrelated to HPV infection.”

We also added the HPV-related data as the Supplementary Table 12.

Supplementary Table 12. HPV status checked by PCR and RT-PCR in samples.

Sample #	HPV-DNA *	HPV-E6 cDNA
25	type 18	undetectable
49	type 16	undetectable
51	type 16	undetectable
56	type 16	undetectable

*HPV-DNA PCR products were confirmed by Sanger’s sequencing.

We tested the presence of HPV DNA in all 50 samples by PCR using genomic DNA, and found 4 cases were positive. However, we did not detect HPV E6 transcript in these 4 cases by RT-PCR using cDNA samples prepared for RNAseq experiments. Three of 4 HPV DNA-positive cases have p53 mutations, typically not related to HPV. This further supported that the mutation signatures in the 50 OSCC cases are not HPV-driven. HPV E6 inactivates p53 through direct protein-protein binding, resulting in deregulation of the cell cycle in HPV-E6 positive cells. Previous reports indicated that the mutation pattern in OSCC tumors without E6 expression is similar to those in HPV-negative tumors^{1,2}, while p53 gene mutations are inversely correlated with HPV E6 expression³. Thus, we can conclude that the mutational signatures observed in the 50 OSCC samples were not driven by HPV.

1. Wichmann G, et al. The role of HPV RNA transcription, immune response-related gene expression and disruptive TP53 mutations in diagnostic and prognostic profiling of head and neck cancer. *Int J Cancer* **137**, 2846-2857

(2015).

2. Braakhuis BJ, *et al.* Genetic patterns in head and neck cancers that contain or lack transcriptionally active human papillomavirus. *J Natl Cancer Inst* **96**, 998-1006 (2004).
3. Westra WH, Taube JM, Poeta ML, Begum S, Sidransky D, Koch WM. Inverse relationship between human papillomavirus-16 infection and disruptive p53 gene mutations in squamous cell carcinoma of the head and neck. *Clin Cancer Res* **14**, 366-369 (2008).

- I have a problem with the labeling of the gains and losses in 1B by gene names. E.g. the FADD gene refers to 11q13 and the critical gene on this amplicon is in my view not FADD but CyclinD1. There is a tremendous amount of functional data on that. So now the readers become confused. I understand the point the authors want to make, but in these losted and gained regions a multitude of genes is present. Hence the regions should be indicated and candidate genes names at max listed in brackets.

Response:

Thanks for the reviewer's suggestion. We notice that *CCND1* is also significantly amplified in our data. Notably, both *FADD* and *CCND1* reported to be important amplified genes in previous head and neck squamous cell carcinoma study by TCGA (Nature 2015). Therefore, we added *CCND1* in Figure 1c and made several modifications in our manuscript and supplementary data.

1. We added *CCND1* in the sentence at line 137 to “Our findings largely recapitulated the profile found in OSCC-TCGA (Supplementary Fig. 4b), which included significantly amplified/deleted regions encompassing genes such as *EGFR*, *FGFR1*, *CCND1*, *FADD*, *FAT1* and *CDKN2A* (Supplementary Table 7 and 8).”
2. *CCND1* was included in Figure 1c of the revised version.
3. We highlight *CCND1* in red in supplementary Table 7.

Supplementary Table 7. Significantly amplified regions from 50 WXS (OSCC-Taiwan)

cytoband	7p11.2	11q13.3	9p24.2	14q11.2	8p11.23
q value	9.43E-09	1.14E-08	0.00055007	0.0015126	0.044871
wide peak boundary	chr7:54636569-55433971	chr11:69062522-70319672	chr9:976836-3454244	chr14:22356070-22983206	chr8:38133758-39349625
genes in wide peak	EGFR SEC61G LANCL2 VSTM2A LOC285878	hsa-mir-548k CCND1 CTTN FGF3 FGF4 PPF1A1 FADD FGF19 SHANK2 MYEOV ANO1 ORAOV1 MIR548K	RFX3 SMARCA2 VLDLR KIAA0020 DMRT2 DMRT3 KCNV2 FLJ35024	[DAD1]	ADAM3A FGFR1 TACC1 ADAM9 WHSC1L1 PLEKHA2 TM2D2 LETM2 HTRA4 ADAM32 ADAM5P RNF5P1 C8orf86

4. We also listed *CCND1* together with *FADD* in supplementary Figure 4.

Supplementary Figure 4. Copy number variations identified in (a) OSCC-Taiwan and (b) OSCC-TCGA. Regions with significantly overrepresented amplification (left) or deletion (right) were listed together with their residual q values; only regions with q values of less than 0.5 are listed. Potential driver genes within these focal alterations are listed, together with the numbers of resident genes in the corresponding peaks (in parentheses). The novel copy number amplified or deleted driver genes (*SMARCA2*, *TRIML1* and *BTNL8*) are marked in red and blue, respectively.

- The authors should explain the X-axes and Y-axes in all Figure legends. Now readers have to check the materials and methods what is displayed.

Response:

Thanks for the reviewer’s suggestion. We have added the description for the X-axes and Y-axes in all figures if applicable.

- In Fig 1 the q-value should be explained, and the VAF that was used indicated. I asked before which VAF was used, and still miss that in the main text. Could be added to the legend of Fig 1.

Response:

We have modified the Fig 1 legend accordingly.

Figure 1. An integrated deep-sequencing approach identifies novel variant features underlying OSCC of a unique demographic origin. Summary data for the 50 OSCC-Taiwan cases. The four blocks correspond to the different types of data attributes. They represent, from top to bottom: (a) Mutation analyses in a series of 50

OSCC samples. The y-axis shows the number of mutation events and the omics data (DNA, exome sequencing; RNA, RNA-Seq; CCP, comprehensive cancer panel), while the x-axis indicates the samples of the individual patients. (b) Heatmap representation of individual genes exhibiting somatic mutations in the 50 OSCC samples. The q values (false discovery rates) represent the significance of each mutated gene, as determined using MutSigCV. (c) Heatmap representation of the copy number variations, compared to those from TCGA and India. SNVs were identified with Mutect, which applies a Bayesian classifier to detect mutations with allelic fractions of 0.1 or less (<10%). For the number of mutation events, the mutation types are broken down by the indicated sequence features. For the mutation (b) and copy number analyses (c), the tables on the right show the percentages of patients with the respective somatic sequence variation or amplification/deletion, as found in the OSCC-Taiwan (TW), OSCC-TCGA (TCGA), and OSCC-India (India) cohorts. na: data not available. (d) the risk exposure and *A3B* deletion genotypes. OSCC patients with the habits of alcohol, betel nut or cigarette are individually marked. For 3 APOBEC3B genotypes, $A3B^{-/-}$, $A3B^{+/-}$, and $A3B^{+/+}$ are shown with full, half and empty squares, respectively.

Reviewer #3 (Remarks to the Author):

The authors have substantially improved the paper and have addressed my prior concerns in the revised manuscript and response information.

Response: Thank you.

REVIEWERS' COMMENTS:

Reviewer #2 (Remarks to the Author):

The paper improves every review round, and the editing helped to improve the readability. The main message of the MS still remains somewhat hidden in the way of writing and presenting the data. I would summarize it as follows: OSCC in Taiwanese show a frequent APOBEC mutational profile, which relates to a A3A-B germline polymorphism in this population that impacts expression of A3A, and is shown to be of clinical relevance in these oral cancers. Please be very careful not suggesting a relation with increased oral cancer risk, as this was not studied.

Few remarks still:

line 41 change APOBEC3A in APOBEC3A expression

lines 48/49 change We thus report the first example of a germline associated prognostic biomarker for OSCC ... We thus report the first example of a germline variant in a specific ethnic population that impacts the mutational profile and prognosis of oral cancer...

line 50 remove biomarkers and. This study is not about biomarkers but germline variants that become of clinical and molecular relevance when oral cancer originate.

line 310 change disease in polymorphism

REVIEWERS' COMMENTS:

Reviewer #2 (Remarks to the Author):

The papers improves every review round, and the editing helped to improve the readability. The main message of the MS still remains somewhat hidden in the way of writing and presenting the data. I would summarize it as follows: OSCC in Taiwanese show a frequent APOBEC mutational profile, which relates to a A3A-B germline polymorphism in this population that impacts expression of A3A, and is shown to be of clinical relevance in these oral cancers. Please be very careful not suggesting a relation with increased oral cancer risk, as this was not studied.

Few remarks still:

line 41 change APOBEC3A in APOBEC3A expression

lines 48/49 change We thus report the first example of a germline associated prognostic biomarker for OSCC ... We thus report the first example of a germline variant in a specific ethnic population that impacts the mutational profile and prognosis of oral cancer...

line 50 remove biomarkers and. This study is not about biomarkers but germline variants that become of clinical and molecular relevance when oral cancer originate.

line 310 change disease in polymorphism

Response:

Thanks for the reviewer's suggestions. We have modified our abstract, and the revised version is as followed:

1. "Oral squamous cell carcinoma (OSCC) is a prominent cancer worldwide, particularly in Taiwan. By integrating omics analyses in 50 matched samples, we uncover in Taiwanese patients a predominant mutation signature associated with cytidine deaminase APOBEC, which correlates with the up-regulation of APOBEC3A expression in the APOBEC3 gene cluster at 22q13. APOBEC3A expression is significantly higher in tumors carrying APOBEC3B-deletion allele(s). High-level APOBEC3A expression is associated with better overall survival, especially among patients carrying APOBEC3B-deletion alleles, as examined in a second cohort (n=188; $p=0.004$). The frequency of APOBEC3B-deletion alleles is about 50% in 143 genotyped OSCC-Taiwan samples (27A3B^{-/-}:89A3B^{+/-}:27A3B^{+/+}), compared to the 5.8% found in 314 OSCC-TCGA samples. We thus report a frequent APOBEC mutational profile, which relates to a APOBEC3B-deletion germline polymorphism in Taiwanese OSCC that impacts expression of

APOBEC3A, and is shown to be of clinical prognostic relevance. Our finding might be recapitulated by genomic studies in other cancer types.”

2. In line 304, “this disease” means the oral cancer. Changing it to “polymorphism” will alter the meaning of this sentence. Therefore, we keep “this disease” in our revised version.